# Anticipating regime shifts by mixing early warning signals from different nodes

Naoki Masuda [1,2] ✉, Kazuyuki Aihara[3] & Neil G. MacLaren [1]

Real systems showing regime shifts, such as ecosystems, are often composed of many dynamical elements interacting on a network. Various early warning signals have been proposed for anticipating regime shifts from observed data. However, it is unclear how one should combine early warning signals from different nodes for better performance. Based on theory of stochastic differential equations, we propose a method to optimize the node set from which to construct an early warning signal. The proposed method takes into account that uncertainty as well as the magnitude of the signal affects its predictive performance, that a large magnitude or small uncertainty of the signal in one situation does not imply the signal's high performance, and that combining early warning signals from different nodes is often but not always beneficial. The method performs well particularly when different nodes are subjected to different amounts of dynamical noise and stress.

Real complex systems often experience sudden and substantial changes, also referred to as regime shifts or tipping events, as environments or the system's internal properties gradually change. Such sudden changes often alter functions of the system, sometimes in an irreversible manner. Tipping phenomena have been used to explain, for example, mass extinctions in ecosystems[1,2], deforestation[3,4], onset of epidemic spreading[5], and progression of mental disorders[6–8] and other diseases[9,10]. No matter whether the system transits from a desirable state to an undesirable one (e.g., species' extinctions, epidemic spreading) or vice versa (e.g., recovery from disease), one is generally interested in anticipating a large regime shift due to a tipping event before it occurs. From dynamical systems points of view, a tipping event probably most famously corresponds to a change in the stability of an equilibrium. Theory suggests that critical slowing down happens near such a tipping point, which one can exploit to construct early warning signals for an impending tipping event[1,11]. Various early warning signals for tipping events, which can be applied to observed data without knowledge of the dynamical system equations, have been proposed[1,5,10–15].

Complex systems whose tipping points we want to anticipate are more often than not composed of dynamical elements interacting on a network. In an ecosystem, animals, plants, or microbial species, for example, are interconnected by mutualistic, prey-predator, and other types of interactions. In a climate system, geographical regions are interconnected by, for example, water and heat transport. Network resilience is a comprehensive approach with which to understand how dynamics on networks respond to perturbations and system failures[16]. For dynamics on networks that possibly show tipping events and are relevant to these applications, it is highly likely that the network structure is complex[17,18] and that, related to this, all the nodes do not emit equally useful early warning signals[19–24]. Specifically, nodes that are about to tip may be emitting more useful early warning signals than other nodes in the same system that are still far from tipping. Evidence in favor of this view is that, as a dynamical system gradually changes, some nodes may tip earlier than others, showing multistage transitions[11,25–29]. Therefore, selecting an appropriate set of sentinel nodes from which one calculates the early warning signal may improve the quality of the signal in addition to saving the cost of observing uninformative nodes.

Several methods for selecting a sentinel node set, denoted by $S$, for constructing early warning signals have been proposed. The dynamical network biomarker theory searches for an $S$ that maximizes a composite index[9,10,30,31]. The index is the average pairwise correlation of the sample of data between the nodes within $S$ multiplied by the

[1]Department of Mathematics, State University of New York at Buffalo, Buffalo, NY 14260-2900, USA. [2]Institute for Artificial Intelligence and Data Science, State University of New York at Buffalo, Buffalo, NY 14260-5030, USA. [3]International Research Center for Neurointelligence, The University of Tokyo Institutes for Advanced Study, The University of Tokyo, Bunkyo City, Japan. ✉e-mail: naokimas@gmail.com

standard deviation of the samples across the nodes in S, which is divided by the average pairwise correlation of the samples between a node in S and a node outside S. Another analytical approach is to identify a multidimensional linear dynamics model near the bifurcation point only from observed data and then use the dominant eigenvector associated with the estimated dynamics to select S (i.e., as the set of the ith nodes such that the ith entry of the dominant eigenvector is the largest in terms of the absolute value)[23]. Note that an earlier study pointed out the usefulness of the dominant eigenvector[19]. Participation of the node in the dominant eigenvector is also used for selecting S in other studies[21,24]. In addition, an eigenvector-based method has also been proposed for ranking the nodes most sensitive to perturbations[32]. A different study numerically showed that nodes with small degrees (i.e., small number of the neighboring nodes) are good performers[20]. A method to determine S based on network control theory also found an advantage of selecting nodes with small degrees[27]. Using the nodes receiving the highest total input from other nodes is also a powerful heuristic for setting S[29]. Simply using the nodes with the highest fluctuations also improves upon other naive methods[29]. Note that the last two methods are often better than using all the nodes in the network as S, and the same may be true for other node selection methods.

Except for the dynamical network biomarker theory, these methods provide ranking of nodes and suggest that we should include the top-ranked nodes in S. It is unclear whether combining the nodes in S improves an early warning signal compared to when the single top node in S is used. Any early warning signal (e.g., variance of the signal, lagged autocorrelation) is noisy because, by design, an early warning signal exploits the information about impending tipping events hidden in noisy observations. If a given early warning signal measured for each ith node is independent for the different nodes in S, then averaging them including the case of a weighted average is expected to produce a better early warning signal owing to the central limit theorem. Quantitatively, the standard deviation of the early warning signal should decrease according to $\propto n^{-1/2}$, where $n$ is the number of nodes in S, and $\propto$ represents "in proportion to". However, nodes in a complex system are interrelated, such as through edges in the case of conventional networks, and the states of different nodes are correlated in general[33–35]. Therefore, it is not a trivial concern whether or not combination of the top-ranked nodes generates a high-quality early warning signal. For example, suppose that S is composed of $n$ nodes close to each other in the network and that the states of these $n$ nodes are hence strongly correlated. In this case, averaging the early warning signals over the $n$ nodes does not help much to reduce their fluctuation because the $n$ early warning signals are similar to each other. We may be then tempted to select $n$ nodes that are far from each other on the network even if some of the $n$ nodes do not provide early warning signals of top quality.

In fact, the aforementioned dynamical network biomarker theory aims to optimize the set S, which the original authors call the dominant group, implicitly resolving this problem[9,10,30,31]. However, the composite index that they propose is heuristic, and why this method works well in medical applications[10,36] and how the effectiveness of their method translates to other applications such as in ecology are elusive. Furthermore, this theory and most other proposals of early warning signals neglect that early warning signals are themselves noisy. Considering fluctuations of early warning signals in their design is necessary for at least two reasons. First, if a candidate early warning signal carries a lot of noise, then that signal may not be useful even if its expected value is sensitive to the approach of the system towards the tipping point. A recent study highlighted interplay of the dominant eigenvector direction of the Jacobian matrix of the dynamics and the primary noise direction that is dragged towards the nodes receiving high dynamical noise[24]. Second, in the aforementioned example of $n$ nodes, let us assume that the expectation of the single-node early

warning signal is the same for all the $n$ nodes. We stated that averaging over the $n$ early warning signals would be beneficial relative to the single-node early warning signals only if the $n$ signals are not strongly correlated with each other. In this situation, the expectation of the averaged early warning signal is the same as that of the single-node early warning signal. The bonus of the averaging is only present in the reduced fluctuation in the averaged as opposed to the single-node early warning signal. To enable such discussion, we need to assess the fluctuation as well as the magnitude of early warning signals.

In the present study, we develop a mathematical framework for node set selection for early warning signals based on stochastic differential equations on networks. By assuming dynamics near the equilibrium and using the variance of the node's state as the early warning signal, we propose an index whose maximization gives an optimal node set for constructing an early warning signal. We demonstrate our method with analytically solvable networks with two or three nodes and with numerically investigated larger networks combined with various dynamics models.

## Results
### Theory
We consider an $N$-dimensional noisy nonlinear dynamical system in continuous time. We regard each of the $N$ dynamical elements as a node and the entire dynamical system as stochastic dynamics on a network with $N$ nodes. We denote by $x_i(t) \in \mathbb{R}$ the state of the $i$th node at time $t \in \mathbb{R}$. We write $\mathbf{x}(t) = (x_1(t), \ldots, x_N(t))^\top$, where $^\top$ represents the transposition. We assume that $\mathbf{x}(t)$ obeys a set of stochastic differential equations in the Itô sense given by

$$d\mathbf{x}(t) = F(\mathbf{x}(t))dt + BdW(t), \tag{1}$$

where $F : \mathbb{R}^N \to \mathbb{R}^N$; $B$ is an $N \times N$ matrix; $W(t)$ represents an $N$-dimensional vector of independent Wiener processes (i.e., white noise).

Assume that the dynamics given by Eq. (1) has an equilibrium in the absence of noise, which we denote by $\mathbf{x}^* = (x_1^*, \ldots, x_n^*)^\top$. By linearizing Eq. (1) around $\mathbf{x}^*$, we obtain a set of linear stochastic differential equations given by

$$d\mathbf{z}(t) = -A\mathbf{z}(t)dt + BdW(t), \tag{2}$$

where $\mathbf{z}(t) = \mathbf{x}(t) - \mathbf{x}^*$, and the $N \times N$ matrix $A$ is the sign-flipped Jacobian matrix of $F$ at $\mathbf{x}(t) = \mathbf{x}^*$. Equation (2) is a multivariate Ornstein-Uhlenbeck (OU) process[24,37], and $\mathbf{x}^*$ is asymptotically stable in the absence of noise if and only if $A$ is positive definite.

The covariance matrix in the equilibrium, corresponding to $\mathbf{z}^* = (0, \ldots, 0)^\top$, is given as the solution to the Lyapunov equation given by

$$AC + CA^\top = BB^\top, \tag{3}$$

where $C = (C_{ij})$ is the $N \times N$ covariance matrix, and $C_{ij}$ represents the covariance between $z_i(t)$ and $z_j(t)$, which is equal to the covariance between $x_i(t)$ and $x_j(t)$, at equilibrium[37,38]. The solution $C$ is unique if the real part of all the eigenvalues of $A$ is positive[38].

Outcomes of the covariance matrix such as the standard deviation and correlation coefficient are often used as early warning signals for noisy multivariate dynamics. In practice, we need to estimate these quantities from samples. Therefore, we consider the sample variance of $x_i(t)$, which is a major early warning signal[1,39–41], and its average over a given node set[29,42], as candidates of early warning signals. The choice of the variance rather than the standard deviation is because of mathematical tractability. We emphasize that the rationale behind averaging the sample variance over nodes is that we may be then able to obtain a less fluctuating early warning signal than that calculated from a single node.

Assume that we observe $L$ samples of $x_i(t)$ from each node $i$ in the given node set $S$. We denote by $\hat{V}_i$ the unbiased sample variance of $x_i(t)$ calculated from the $L$ samples. Let us consider the average of $\hat{V}_i$ over the $n = |S|$ nodes in set $S$ as an early warning signal. Without loss of generality, we assume that $S = \{1, 2, ..., n\}$, where $n \leq N$. We denote this average by

$$\hat{V}_S = \frac{1}{n}\sum_{i=1}^{n}\hat{V}_i. \tag{4}$$

By assuming that $z_{i,1}, ..., z_{i,L}$ are i.i.d. and using $E[z_{i,\ell}] = 0$, we obtain

$$E[\hat{V}_S] = \frac{1}{n}\sum_{i=1}^{n}E[\hat{V}_i] = \frac{1}{n}\sum_{i=1}^{n}C_{ii} = \frac{1}{n}\,\mathrm{Tr}\,(\overline{C}) = \frac{1}{n}\sum_{i=1}^{n}\lambda_i, \tag{5}$$

where $E$ denotes the expectation, Tr denotes the trace, $\overline{C}$ is the leading principal minor of order $n$ of $C$ (i.e., the submatrix of $C$ composed of $C_{ij}$ with $i, j \in \{1, ..., n\}$), and $\lambda_i$ is the $i$th eigenvalue of $\overline{C}$. As shown in Supplementary Note 1, we also obtain the variance of $\hat{V}_S$, denoted by $\mathrm{var}[\hat{V}_S]$, as follows:

$$\begin{aligned}\mathrm{var}[\hat{V}_S] &= \frac{2}{n^2(L-1)}\sum_{i=1}^{n}\sum_{j=1}^{n}(C_{ij})^2 \\ &= \frac{2}{n^2(L-1)}\,\mathrm{Tr}\,(\overline{C}^2) \\ &= \frac{2}{n^2(L-1)}\sum_{i=1}^{n}\lambda_i^2. \end{aligned} \tag{6}$$

The coefficient of variation (CV), i.e., the standard deviation divided by the mean, of $\hat{V}_S$, which quantifies the relative uncertainty in the estimation of $\hat{V}_S$, is given by

$$CV = \sqrt{\frac{2}{L-1}}\frac{\sqrt{\sum_{i=1}^{n}\lambda_i^2}}{\sum_{i=1}^{n}\lambda_i}. \tag{7}$$

Equation (7) has a couple of implications. First, the CV decays according to $\propto L^{-1/2}$ as the number of samples, $L$, increases, regardless of the node set $S$. This scaling corresponds to the central limit theorem. Second, in the case of a single node, $CV = \sqrt{2/(L-1)}$ regardless of the node. Therefore, a large value of the sample covariance, $\hat{V}_i$, or its expectation, $E[\hat{V}_i]$, at a node $i$ compared to at other nodes does not imply that $\hat{V}_i$ is better than $\hat{V}_j$'s (with $j \neq i$) as an early warning signal. A large signal carries a proportionately large amount of noise. This property also holds true when one uses the sample standard deviation, instead of the sample variance, of single nodes as early warning signal. Third, the CV with $n \geq 2$ is always smaller than the CV with $n = 1$ (i.e., $\sqrt{2/(L-1)}$) unless all but one eigenvalues of $\overline{C}$ are equal to 0. Note that this result holds true owing to $\lambda_i \geq 0\ \forall i$, which follows from the fact that $\overline{C}$ is a covariance matrix and therefore positive semidefinite. This result motivates us to use the average of $\hat{V}_i$ over nodes as opposed to $\hat{V}_i$ at a single node with the aim of improving the performance of the early warning signal. We will investigate this possibility in the following sections.

## Coupled nonlinear dynamics on networks with two or three nodes

In this section, we analyze coupled nonlinear dynamics on networks with $N = 2$ and $N = 3$ nodes. Through this analysis, we highlight relevance of multistage transitions, impact of averaging $\hat{V}_i$ over nodes, heterogeneous amount of dynamical noise given to different nodes, and uncertainty of early warning signals, among other things. We then propose a measure of the quality of early warning signals in the form of

$\hat{V}_S$ and a method to select an optimal node set for constructing early warning signals.

**Two nodes connected by a directed edge.** We consider a network composed of two nodes and a directed edge of weight $w(\geq 0)$; see Fig. 1a for a schematic. We assume that node 1 influences node 2 but not vice versa. We also assume that, as a bifurcation parameter, denoted by $r$, gradually increases, node 1 undergoes a saddle-node bifurcation and that node 2 also undergoes a saddle-node bifurcation either almost at the same time as node 1 or after $r$ has further increased. The model is given by

$$dx_1(t) = f(x_1(t), r)dt + \sigma_1 dW_1(t), \tag{8}$$

$$dx_2(t) = [f(x_2(t), r - \Delta r) + w(x_1(t) + 1)]dt + \sigma_2 dW_2(t), \tag{9}$$

where $\Delta r(\geq 0)$ is a constant, $\sigma_1$ and $\sigma_2$ are the intensities of dynamical noise applied to nodes 1 and 2, respectively, and $f(x)$ satisfies the following conditions. First, we assume $f(x, r) = r + x^2$ when $r \leq 0$ and $x \leq \sqrt{-r} + \Delta x$, where $\Delta x(>0)$ is a small constant. This condition guarantees that, in the absence of coupling and dynamical noise, Eqs. (8) and (9) are both the topological normal form of the saddle-node bifurcation[43]. In other words, $dx/dt = f(x, r)$ with $r < 0$ has a stable equilibrium $x^* = -\sqrt{-r}$ and an unstable equilibrium $x^* = \sqrt{-r}$, which collide at $x^* = 0$ when $r = 0$. In Fig. 2, we show an example bifurcation diagram of single-node deterministic dynamics given by $dx/dt = f(x, r)$. If $\sigma_1 = 0$, then $x_1(t)$ undergoes a saddle-node bifurcation at $r = 0$ as $r$ increases starting with a negative value. If $\sigma_2 = 0$ and $w = 0$, then $x_2(t)$ undergoes a saddle-node bifurcation at $r = \Delta r$. Second, we assume that $f(x, r)$ is continuous in terms of $x$ and $r$ for simplicity. Third, we assume that $f(c, r) = 0$ for $\forall r \geq 0$ for a unique positive value of $c$, which is larger than $\Delta x$. This implies that, in the absence of noise, $x = c$ is the unique

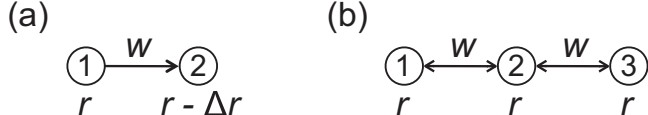

(a) ① $\xrightarrow{w}$ ②
 $r$ $r - \Delta r$

(b) ① $\xleftarrow{w}$ ② $\xleftarrow{w}$ ③
 $r$ $r$ $r$

**Fig. 1 | Schematic of two networks. a** A network with $N = 2$ nodes connected by a directed edge. **b** A symmetric chain network with $N = 3$ nodes. The coupling strength is denoted by $w$. The stress given to each node is either $r$ or $r - \Delta r$.

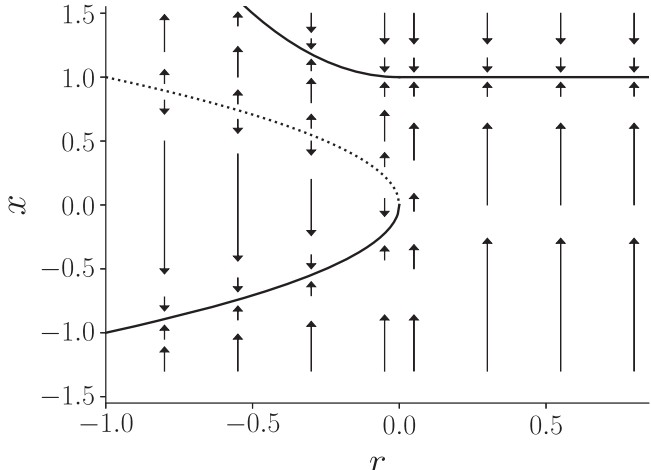

**Fig. 2 | An example bifurcation diagram of single-node dynamics given by Eq. (8) without dynamical noise.** The solid and dashed lines represent stable and unstable equilibria, respectively.

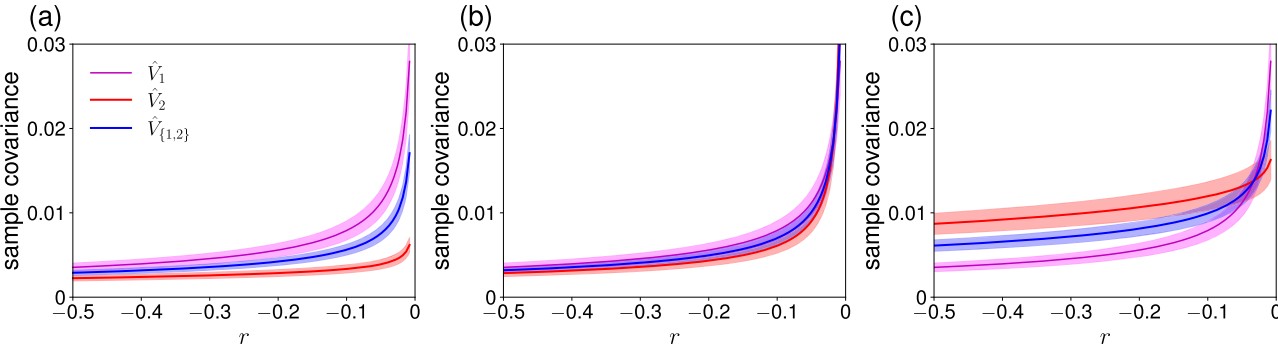

**Fig. 3 | Early warning signals with different node sets in a network with $N = 2$ nodes connected by a directed edge.** The solid lines represent the mean. The shaded regions represent the standard deviation. We set $w = 0.5$, $\sigma_1 = 0.1$, and $L = 100$. **a** $(\sigma_2, \Delta r) = (0.1, 1)$. **b** $(\sigma_2, \Delta r) = (0.1, 0.5)$. **c** $(\sigma_2, \Delta r) = (0.2, 1)$.

stable equilibrium after a node undergoes a saddle-node bifurcation as $r$ gradually increases. This assumption in combination with the continuity assumption for $f(x, r)$ also implies that the stable equilibrium apart from $x^* = -\sqrt{-r}$ persists for some $r < 0$ although its position changes from $x = c$ in general. Therefore, there are two stable equilibria at least in some range of $x < 0$ near $x = 0$, as shown in Fig. 2.

We assume that $w > 0$ and consider this dynamical system in the range of bifurcation parameter $r \in [-1, 0)$, with $(x_1, x_2)$ satisfying $-1 \le x_1, x_2 < 0$ in the absence of dynamical noise. In other words, we assume that $x_1$ and $x_2$ are both near the lower stable equilibrium in Fig. 2. In this situation, the input that node 2 receives from node 1, i.e., $w(x_1 + 1)$, is positive. To prevent node 2 from transiting from its lower to the upper state through a saddle-node bifurcation earlier than node 1 when $r$ gradually increases, we assume that $w \le \Delta r$, which guarantees that $-\Delta r + w(x_1 + 1) \le 0$. Let us gradually increase $r$ starting with $r = -1$. Then, $x_1$ jumps from 0 to $c$ at $r = 0$ via a saddle-node bifurcation. We distinguish between two cases depending on the change in $x_2$ right after the transition of node 1 from $x_1 = 0$ to $x_1 = c$. If $-\Delta r + w(c + 1) > 0$, then $x_2$ transits from $x_2 = 0$ to $x_2 = c$ immediately after $x_1$ does without a further increase in $r$ (i.e., at $r = 0$). Otherwise, $x_2$ does not transit at $r = 0$, and $x_2$ does so at a positive value of $r$ if we further increase $r$. The latter case is a multistage transition[28,29].

The lower equilibrium of the coupled dynamical system, which exists if and only if $r \le 0$ and is locally stable when $r < 0$, is given by

$$\mathbf{x}^* = \begin{pmatrix} x_1^* \\ x_2^* \end{pmatrix} = \begin{pmatrix} -\sqrt{-r} \\ -\sqrt{-r + \Delta r + w(\sqrt{-r} - 1)} \end{pmatrix}. \quad (10)$$

The linearized dynamics around $\mathbf{x}^*$ is given by Eq. (2) with

$$A = \begin{pmatrix} -2x_1^* & 0 \\ -w & -2x_2^* \end{pmatrix}. \quad (11)$$

By combining Eqs. (3) and (11), we obtain

$$C_{11} = -\frac{\sigma_1^2}{4x_1^*}, \quad (12)$$

$$C_{12} = \frac{\sigma_1^2 w}{8x_1^*(x_1^* + x_2^*)}, \quad (13)$$

$$C_{22} = -\frac{\sigma_1^2 w^2}{16x_1^* x_2^*(x_1^* + x_2^*)} - \frac{\sigma_2^2}{4x_2^*}. \quad (14)$$

Note that $x_1^*$ and $x_2^*$ are negative such that $C_{11}$, $C_{12}$, and $C_{22}$ are positive.

Here we recall that $E$ denotes the expectation, and we let std denote the standard deviation. Because $x_1^* = 0^-$ as $r \to 0^-$, all of $C_{11}$, $C_{12}$,

and $C_{22}$ diverge as $r \to 0^-$. Therefore, $E[\hat{V}_1]$, $E[\hat{V}_2]$, and $E[\hat{V}_{\{1,2\}}]$, which are equal to $C_{11}$, $C_{22}$, and $(C_{11} + C_{22})/2$, respectively, all diverge as $r \to 0^-$ according to $\propto (-r)^{-1/2}$. However, we argue that this fact does not immediately imply that $\hat{V}_1$, $\hat{V}_2$, or $\hat{V}_{\{1,2\}}$ is a high-quality early warning signal for the saddle-node bifurcation occurring at $r = 0$ because $\text{std}[\hat{V}_1]$, $\text{std}[\hat{V}_2]$, and $\text{std}[\hat{V}_{\{1,2\}}]$, which are proportional to $C_{11}$, $\sqrt{C_{11}^2 + 2C_{12}^2 + C_{22}^2}$, and $C_{22}$, respectively, also diverge.

To investigate the quality of $\hat{V}_1$, $\hat{V}_2$, and $\hat{V}_{\{1,2\}}$ as early warning signals, we set $w = 0.5$, $\sigma_1 = 0.1$, $L = 100$, and used three pairs of $\Delta r$ and $\sigma_2$ values, which we refer to as scenarios. In Fig. 3, we show $E[\hat{V}_1]$, $E[\hat{V}_2]$, $E[\hat{V}_{\{1,2\}}]$, $\text{std}[\hat{V}_1]$, $\text{std}[\hat{V}_2]$, and $\text{std}[\hat{V}_{\{1,2\}}]$ as we gradually increase $r$ under the three scenarios. In Fig. 3a, we show the results for the first scenario, for which we set $\Delta r = 1$ and $\sigma_2 = \sigma_1 = 0.1$. The solid lines represent the expected value of each early warning signal. The shaded area represents the mean ± standard deviation. With these parameter values, when node 1 transits from $x_1 = 0$ to $x_1 = c$ at $r = 0$, node 2 is still far from the bifurcation point. This is because the input from node 1 to node 2, which is equal to $w(x_1 + 1) \approx 0.5$, is substantially smaller than $-\Delta r(=1)$ such that $x_2$ approximately obeys $dx_2 = (-0.5 + x^2) dt + \sigma_2 dW_2(t)$ near $r = 0$. Right after node 1 transits from $x_1 = 0$ to $x_1 = c$ at $r = 0$, the input from node 1 to node 2 jumps to $w(x_1 + 1) \approx 0.5(c + 1)$. Therefore, a multistage transition in which nodes 1 and 2 transit from their lower to the upper state at different $r$ values occurs if $0.5(c + 1) < 1$, i.e., $0 < c < 1$. Otherwise (i.e., $c \ge 1$), node 2 also transits from its lower to the upper state at $r = 0$. Regardless of whether or not a multistage transition occurs, Fig. 3a indicates that $E[\hat{V}_1]$ is more sensitive to increases in $r$ than $E[\hat{V}_2]$ and $E[\hat{V}_{\{1,2\}}]$, and that $\text{std}[\hat{V}_1]$ is larger than $\text{std}[\hat{V}_2]$ and $\text{std}[\hat{V}_{\{1,2\}}]$. Therefore, it is not obvious which of $\hat{V}_1$, $\hat{V}_2$, or $\hat{V}_{\{1,2\}}$ is a better early warning signal.

To rank these different candidates of early warning signals, we define the following index. We measure each early warning signal's mean and standard deviation at $r = -0.3$ and $r = -0.1$. We do so based on the premise that it is necessary to measure signals at least at two $r$ values to estimate how responsive the signal is to the change in the environment, i.e., $r$. As we explained in Supplementary Note 1, $\hat{V}_1$, $\hat{V}_2$, and $\hat{V}_{\{1,2\}}$ obey a normal distribution at equilibrium. Therefore, we measure a distance, denoted by $d$, between the normal distribution at $r = -0.3$ and that at $r = -0.1$ for each early warning signal. If $d$ is large, it is relatively easy to tell the increase in the signal with relatively small uncertainty as $r$ increases from $-0.3$ to $-0.1$, approaching a tipping point. Therefore, we propose that an early warning signal attaining a larger $d$ value is better. Although there are various measures of the separability of two normal distributions[44], inspired by the $t$-statistic, we define

$$d = \frac{|\mu_1 - \mu_2|}{\sqrt{\text{var}_1 + \text{var}_2}}, \quad (15)$$

where $\mu_1$ and $\mathrm{var}_1$ are the mean and variance of the normal distribution at $r = -0.3$, and $\mu_2$ and $\mathrm{var}_2$ are those at $r = -0.1$. We show in Supplementary Note 2 that the results shown in the remainder of this section are robust with respect to the choice of the two $r$ values for calculating $d$. Note that $d$ can be small even if the mean of the early warning signal grows by a large amount between the two $r$ values. This is the case when $\mathrm{var}_1$ or $\mathrm{var}_2$ is large, i.e., when the uncertainty of the signal is large. In this situation, tracking the early warning signal is relatively uninformative. In contrast, even if the absolute magnitude of the increase in the signal is small (i.e., small $|\mu_1 - \mu_2|$), the signal provides a good early warning if the uncertainty of the signal is small.

We have found that $d(\hat{V}_1) = 2.58$, $d(\hat{V}_2) = 1.27$, and $d(\hat{V}_{\{1,2\}}) = 2.78$. Therefore, we suggest that $\hat{V}_1$ is a better early warning signal than $\hat{V}_2$. We emphasize that it is not because $\hat{V}_1$ is larger than $\hat{V}_2$ but because $\hat{V}_1$ responds more strongly to an increase in $r$ than $\hat{V}_2$ does with relatively small uncertainty. Furthermore, $\hat{V}_{\{1,2\}}$, i.e., the average of $\hat{V}_1$ and $\hat{V}_2$, realizes a larger value of $d$ than $\hat{V}_1$. Therefore, we suggest that $\hat{V}_{\{1,2\}}$ is a better early warning signal than $\hat{V}_1$ and $\hat{V}_2$. This is intuitively because the averaging cancels out noise in the two signals, while $\hat{V}_1$ and $\hat{V}_2$ are not independent of each other due to the edge between nodes 1 and 2. We note that $E[\hat{V}_{\{1,2\}}]$ is smaller than $E[\hat{V}_1]$ (see Fig. 3a), challenging a natural idea of using single nodes or node sets with the largest variance as sentinel nodes.

We show in Fig. 3b the mean and standard deviation of the signals for the second scenario; we changed $\Delta r$ from 1 to 0.5. In this scenario, $x_2(t)$ always transits from its lower to the upper state immediately after node 1 does so at $r = 0$. This is because, when $r \to 0^-$, Eq. (8) implies that $x_1(t) \to 0^-$ if we ignore the noise term, and $r - \Delta r + w(x_1(t) + 1) \approx 0$ combined with Eq. (9) implies that $x_2(t) \to 0^-$. Therefore, even with a small $c$ value, both nodes 1 and 2 sequentially transit from its lower to the upper state at $r = 0$. Figure 3b indicates that $E[\hat{V}_2]$ is responsive to increases in $r$ to an extent similar to $E[\hat{V}_1]$ is. The $d$ values confirm this, i.e., $d(\hat{V}_1) = 2.58$, $d(\hat{V}_2) = 2.50$, and $d(\hat{V}_{\{1,2\}}) = 3.40$. Because the quality of $\hat{V}_2$ is much better than in the first scenario, the average signal, $\hat{V}_{\{1,2\}}$, is substantially better than $\hat{V}_1$ in terms of $d$ in the present second scenario, while $\hat{V}_{\{1,2\}}$ was only marginally better than $\hat{V}_1$ in the first scenario.

In addition to different amounts of constant stress as parametrized by $\Delta r$, different nodes may be subject to different amounts of dynamical noise. We show in Fig. 3c the results for the third scenario, in which we set $\Delta r = 1$ and $\sigma_2 = 0.2$. In this scenario, the saddle-node bifurcation for node 2 is delayed such that a multistage transition may occur, as in the first scenario. The difference to the first scenario is that, in the present scenario, node 2 receives a larger dynamical noise than node 1. Therefore, $\hat{V}_2$ is noisier than $\hat{V}_1$ and $\hat{V}_{\{1,2\}}$, which is apparent in Fig. 3c. If we only measure the early warning signals at one $r$ value (e.g., $r = -0.3$ or $r = -0.1$), then one may be tempted to regard that $\hat{V}_2$ is a better early warning signal than $\hat{V}_1$ and $\hat{V}_{\{1,2\}}$ because $E[\hat{V}_2]$ is larger than $E[\hat{V}_1]$ and $E[\hat{V}_{\{1,2\}}]$. However, this conclusion is erroneous for two reasons. First, $\mathrm{std}[\hat{V}_2]$ is larger than $\mathrm{std}[\hat{V}_1]$ and $\mathrm{std}[\hat{V}_{\{1,2\}}]$. Second, $\hat{V}_2$ is not much sensitive to the change in $r$, as Fig. 3c shows. In fact, we obtain $d(\hat{V}_1) = 2.58$, $d(\hat{V}_2) = 0.97$, and $d(\hat{V}_{\{1,2\}}) = 2.12$. This result suggests that $\hat{V}_1$ is a better signal than $\hat{V}_2$ and $\hat{V}_{\{1,2\}}$. In this case, taking the average of $\hat{V}_1$ and $\hat{V}_2$ does not help because $\hat{V}_2$ is too poor. Lessons from the analysis of the present scenario are that (i) a large signal value does not necessarily imply a better early warning signal and that (ii) it is sometimes better to exclude some nodes from the calculation of the early warning signal if those nodes are either only marginally responsive to the change in the bifurcation parameter or they carry larger fluctuations than other nodes.

**Chain with three nodes.** In this section, we consider three nodes of identical nonlinear dynamics, except possible differences in the noise strength, coupled as an undirected chain network with $N = 3$ nodes. We show a schematic of the network in Fig. 1b. Specifically, we consider the

set of stochastic differential equations given by

$$dx_1 = [f(x_1) + w(x_2 + 1)]dt + \sigma_1 dW_1, \tag{16}$$

$$dx_2 = [f(x_2) + w(x_1 + 1) + w(x_3 + 1)]dt + \sigma_2 dW_2, \tag{17}$$

$$dx_3 = [f(x_3) + w(x_2 + 1)]dt + \sigma_3 dW_3, \tag{18}$$

where $f(x)$ is given in the "Two nodes connected by a directed edge" section. If $c$ is small enough and $\sigma_1 = \sigma_2 = \sigma_3$, node 2 transits from its lower to the upper state earlier than nodes 1 and 3 as $r(\geq -1)$ increases from a negative value because node 2 receives positive input from nodes 1 and 3 while node 1 and 3 each receive input solely from node 2.

When $-1 \leq r \leq 0$, the equilibrium in the absence of noise satisfying $x_1^*, x_2^*, x_3^* < 0$ is a solution of

$$r + (x_1^*)^2 + w(x_2^* + 1) = 0, \tag{19}$$

$$r + (x_2^*)^2 + 2w(x_1^* + 1) = 0, \tag{20}$$

and $x_3^* = x_1^*$. If we ignore the dynamical noise, the first saddle-node bifurcation, which is the transition of node 2 from its lower to the upper state, occurs at $r = r_c$, where $r_c$ satisfies $0 > r_c > r_c'' \equiv 2w[-(w+1) + \sqrt{w(w+1)}]$, as $r$ gradually increases from $r = -1$. At $r = r_c$, the two parabolas given by Eqs. (19) and (20) are tangent to each other. See Supplementary Note 3 for the derivation including that of the uniqueness of the stable solution satisfying $x_1^*(= x_3^*) < 0$ and $x_2^* < 0$ when $r \leq r_c$. We set $w = 0.05$, which leads to $r_c \approx r_c'' \approx -0.082$. Therefore, we calculate the $d$ values at $r = -0.3$ and $-0.1$ as we did in the "Two nodes connected by a directed edge" section. We show in Supplementary Note 2 that the following results are reasonably robust against variation in the two $r$ values.

We obtain the mean and standard deviation of the early warning signals, and then $d$, as follows. Equations (16), (17), and (18) lead to

$$A = \begin{pmatrix} -2x_1^* & -w & 0 \\ -w & -2x_2^* & -w \\ 0 & -w & -2x_1^* \end{pmatrix}. \tag{21}$$

To facilitate further analyses, we assume $\sigma_1 = \sigma_3$. Then, by substituting Eq. (21) and $B = \mathrm{diag}(\sigma_1, \sigma_2, \sigma_1)$, which is by definition the diagonal matrix whose diagonal entries are $\sigma_1$, $\sigma_2$, and $\sigma_1$ in this order, into Eq. (3), we obtain

$$C_{11} = C_{33} = \frac{-4x_1^* x_2^* (x_1^* + x_2^*)\sigma_1^2 + w^2[(2x_1^* + x_2^*)\sigma_1^2 - x_1^* \sigma_2^2]}{8x_1^*(2x_1^* x_2^* - w^2)(x_1^* + x_2^*)}, \tag{22}$$

$$C_{12} = C_{23} = \frac{w(x_2^* \sigma_1^2 + x_1^* \sigma_2^2)}{4(2x_1^* x_2^* - w^2)(x_1^* + x_2^*)}, \tag{23}$$

$$C_{13} = -\frac{w^2(x_2^* \sigma_1^2 + x_1^* \sigma_2^2)}{8x_1^*(2x_1^* x_2^* - w^2)(x_1^* + x_2^*)}, \tag{24}$$

$$C_{22} = \frac{-2x_1^* x_2^*(x_1^* + x_2^*)\sigma_2^2 + w^2 x_2^*(-\sigma_1^2 + \sigma_2^2)}{4x_2^*(2x_1^* x_2^* - w^2)(x_1^* + x_2^*)}. \tag{25}$$

By substituting Eqs. (22)–(25) into Eqs. (5) and (6), we obtain the mean and standard deviation of each early warning signal for any node set $S$. Because $\hat{V}_1$ and $\hat{V}_3$ obey the same normal distribution, we obtain $E[\hat{V}_{\{1,3\}}] = E[(\hat{V}_1 + \hat{V}_3)/2] = E[\hat{V}_1] = E[\hat{V}_3]$. We also obtain

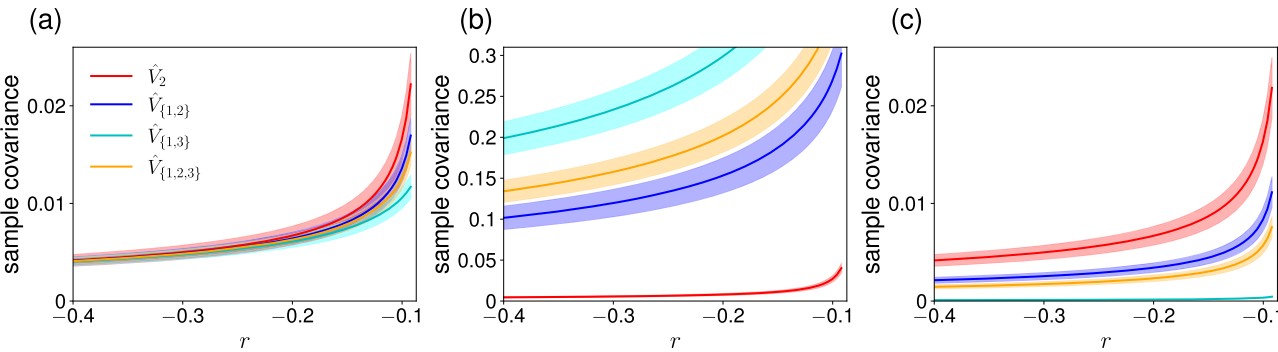

**Fig. 4 | Early warning signals with different node sets in the undirected chain network with $N = 3$ nodes.** We set $w = 0.05$, $\sigma_2 = 0.1$, and $L = 100$. **a** $\sigma_1 = 0.1$. **b** $\sigma_1 = 0.7$. **c** $\sigma_1 = 0.015$.

$\mathrm{std}[\hat{V}_{\{1,3\}}] = (C_{11} + 2C_{13} + C_{33})/[2(L-1)] = \mathrm{std}[\hat{V}_1] \times \frac{1 + Corr}{2}$, where $Corr$ is the Pearson correlation coefficient between $x_1(t)$ and $x_3(t)$ in the equilibrium. Therefore, $\mathrm{std}[\hat{V}_{\{1,3\}}]$ is smaller than $\mathrm{std}[\hat{V}_1]$ unless $x_1(t)$ and $x_3(t)$ are perfectly correlated, which does not happen because they are subjected to independent Wiener processes $dW_1(t)$ and $dW_3(t)$. Therefore, $\hat{V}_{\{1,3\}}$ is always better than $\hat{V}_1$ and $\hat{V}_3$ for this model. In addition, due to the assumed symmetry between nodes 1 and 3, $S = \{1,2\}$, $\{1,3\}$, and $\{1,2,3\}$ exhaust all possibilities of combining multiple nodes' signals into one early warning signal. Therefore, it suffices to compare the performance of $\hat{V}_2$, $\hat{V}_{\{1,2\}}$, $\hat{V}_{\{1,3\}}$, and $\hat{V}_{\mathrm{all}} \equiv (\hat{V}_1 + \hat{V}_2 + \hat{V}_3)/3$, i.e., $\hat{V}_S$ with $S = \{1,2,3\}$.

We set $\sigma_2 = 0.1$ and $L = 100$, and consider three values of $\sigma_1$. In the first scenario, we set $\sigma_1 = \sigma_2 = 0.1$. For this case, we show the mean and standard deviation of $\hat{V}_2$, $\hat{V}_{\{1,2\}}$, $\hat{V}_{\{1,3\}}$, and $\hat{V}_{\mathrm{all}}$ as a function of $r$ in Fig. 4a. Note that the result for $\hat{V}_1$ is the same as that for $\hat{V}_{\{1,3\}}$ except that $\mathrm{std}[\hat{V}_1]$ is $2/(1 + Corr)(>1)$ times larger than $\mathrm{std}[\hat{V}_{\{1,3\}}]$. It is not straightforward to tell from Fig. 4a which signal is better than others. However, we expect that $\hat{V}_2$ provides a better early warning signal than $\hat{V}_1$ because the transition of node 2 from the lower to the upper state as $r$ increases from $r \approx -0.5$ is more impending than the transition of nodes 1 and 3. In fact, we obtain $d(\hat{V}_1) = 3.52$, $d(\hat{V}_2) = 4.72$, $d(\hat{V}_{\{1,2\}}) = 5.84$, $d(\hat{V}_{\{1,3\}}) = 4.97$, and $d(\hat{V}_{\mathrm{all}}) = 6.77$, verifying this prediction. Furthermore, it is the best to use $\hat{V}_{\mathrm{all}}$, i.e., the average of the sample covariance over all nodes.

In the second scenario, we set $\sigma_1 = 0.7$ such that nodes 1 and 3 receive larger dynamical noise than node 2. We show the behavior of the early warning signals in this case in Fig. 4b. The figure indicates that the mean magnitude of any signals including $\hat{V}_1$ (i.e., $\hat{V}_{1,2}$, $\hat{V}_{1,3}$, and $\hat{V}_{\mathrm{all}}$) is much larger than that of $\hat{V}_2$. However, this phenomenon does not imply that it is better to use a signal including $\hat{V}_1$ because the enhanced signal size owes to the increased amount of dynamical noise. We obtain $d(\hat{V}_1) = 3.49$, $d(\hat{V}_2) = 5.49$, $d(\hat{V}_{\{1,2\}}) = 3.74$, $d(\hat{V}_{\{1,3\}}) = 4.93$, and $d(\hat{V}_{\mathrm{all}}) = 5.10$. Therefore, we suggest that we should only use node 2 as early warning signal under the present scenario because $\hat{V}_2$ is the least affected by the dynamical noise. Note that the magnitude of $\hat{V}_2$ in the present scenario is similar to that in the first scenario (see Fig. 4a).

In the third scenario, we set $\sigma_1 = 0.015$ such that node 2 receives large dynamical noise than nodes 1 and 3. We show the behavior of the early warning signals in this case in Fig. 4c. We find that $E[\hat{V}_2]$ is much larger than $E[\hat{V}_{\{1,3\}}] (= E[\hat{V}_1])$, which does not use $\hat{V}_2$. Note that, the magnitude of $\hat{V}_2$ is similar to that in the last two scenarios because we kept $\sigma_2 = 0.1$ in all the three scenarios. We obtain $d(\hat{V}_1) = 4.50$, $d(\hat{V}_2) = 4.69$, $d(\hat{V}_{\{1,2\}}) = 4.77$, $d(\hat{V}_{\{1,3\}}) = 6.08$, and $d(\hat{V}_{\mathrm{all}}) = 4.84$. Therefore, we suggest that $\hat{V}_{\{1,3\}}$ is the best early warning signal under the present scenario. In the second and third scenarios, the best early warning signal turns out to be the one having the smallest mean magnitude at both $r = -0.3$ and $r = -0.1$.

## Numerical results for larger networks and various dynamical systems

We propose to use the node set $S$ maximizing $d$ as the sentinel node set from which we calculate the early warning signal, $\hat{V}_S$. In the networks with two or three nodes analyzed in the "Coupled nonlinear dynamics on networks with two or three nodes" section, the maximizers of $d$ are composed of nodes with small dynamical noise (i.e., small $\sigma_i$). However, in networks with larger numbers of nodes, it may not be the best to select nodes with small dynamical noise. It is because $x_i$'s of these nodes may be strongly correlated with each other, which typically occurs when these nodes are closely located in the network. In this case, averaging $\hat{V}_i$ over these nodes, which yields $\hat{V}_S$, does not help much to reduce the fluctuation, potentially making this $\hat{V}_S$ a poor early warning signal. Therefore, to test the performance of our method to select the sentinel node set in networks with larger numbers of nodes, we carry out numerical simulations with different networks and dynamical systems in this section.

**Setup for numerical simulations.** We use six undirected and unweighted networks with $N$ nodes and denote a network's adjacency matrix by $(w_{ij})$, with $w_{ii} = 0$ and $w_{ij} = w_{ji} \in \{0, 1\}$ $\forall$ $i, j \in \{1, \ldots, N\}$.

Our theory requires the covariance matrix, $C$, or at least its eigenvalues, estimated at two values of the bifurcation parameter. In the previous sections, we theoretically calculated $C$ by linearizing the original dynamics around the equilibrium and solving the Lyapunov equation given by Eq. (3). However, to know matrix $A$ in Eq. (3), we need to know the term driving the dynamics of the node, i.e., the derivative of $F$ in Eq. (1), which depends on the single node's intrinsic dynamics and form of the coupling between nodes. However, we usually do not have access to such detailed information given empirical data, while estimating them from observed data is an active research field (e.g., ref. 23). Therefore, we use the sample covariance matrix as an estimator of $C$. This choice is for simplicity, and there are better estimators of the covariance matrix that are useful especially when the number of samples, $L$, is small relative to the number of variables, $N$[45]; we will discuss this limitation of the present study in the "Discussion" section.

**Results for the coupled double-well model.** We first consider a coupled double-well model on networks with dynamical noise given by

$$dx_i = \left[ -(x_i - r_1)(x_i - r_2)(x_i - r_3) + D \sum_{j=1}^{N} w_{ij} x_j + u_i \right] dt + \sigma_i dW_i, \quad i \in \{1, \ldots, N\}. \tag{26}$$

Equation (26) represents dynamics of species abundance[25] or climates in interconnected regions[26]. Parameters $r_1, r_2$, and $r_3$ control the

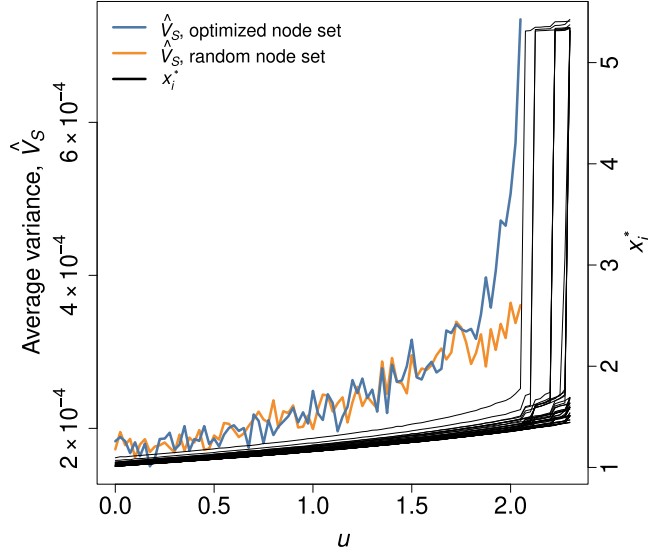

**Fig. 5 | Early warning signal, $\hat{V}_S$, for the maximizer of $d$ (blue line) and a set of nodes selected uniformly at random (orange line).** Both sentinel node sets are composed of $n = 5$ nodes. We also show $x_i^*$ for each node at each value of $u$. We used the coupled double-well dynamics on a BA network with $N = 50$ nodes and gradually increased $u$.

location of the equilibria and satisfy $r_1 < r_2 < r_3$; $D(\geq 0)$ is the strength of coupling between nodes, and $u_i$ is a constant stress given to the $i$th node, equivalent to $-\Delta r$ in Eq. (9). When $D$ or $u_i \; \forall \, i$ is sufficiently small, $\mathbf{x} = (x_1, \ldots, x_N)^{\top}$ in which all the nodes are in their lower state near $r_1$ is the unique stable equilibrium. In contrast, when $D$ or $u_i \; \forall \, i$ is sufficiently large, $\mathbf{x}$ in which all the nodes are in their upper state near $r_3$ is the unique stable equilibrium. In between, we initially set all the nodes in their lower state at the start of each simulation, and as one gradually increases $D$ or $u_i$, all nodes may flip to the upper stable state at once or in multiple stages[28,29].

For this system, we numerically assess the performance of the variance of a single node or its average over the nodes in set $S$ as early warning signal to anticipate the first transition from the lower state to the upper state. We use either $u_i$ or $D$ as the bifurcation parameter in a given sequence of simulations. If the bifurcation parameter is $u_i$, we started with $D = 0.05$ and $u_i = u = 0 \; \forall \, i$ and gradually increased $u$ until the first transition occurs (see the Methods section for the precise definition of the transition and parameter values for the numerical simulations). We computed the covariance matrix at reasonably separated two values of $u$, denoted by $u^{(1)}$ and $u^{(2)}$ (see the Methods section for details). At each of $u = u^{(1)}$ and $u = u^{(2)}$, we collected $L = 100$ samples of each $x_i$, $i \in S$ and calculated $d$ using Eq. (15). As a demonstration, we show how noisy early warning signals gradually increase as we gradually increase $u$ in one series of simulations in Fig. 5. When we select $n = 5$ nodes uniformly at random, the sample covariance averaged over the nodes in $S$ (i.e., $\hat{V}_S$) gradually increases until the first node transits from its lower to the upper state (see the orange line). When we use the set of $n = 5$ nodes maximizing $d$, the sensitivity of $\hat{V}_S$ to the increase in $u$ is notably larger near the first tipping point (see the blue line). Therefore, the maximizer of $d$ provides an apparently better early warning signal than a uniformly randomly selected node set.

We assessed the performance of the sample covariance averaged over the nodes in $S$ using the Kendall's $\tau$, which is conventionally used[13]. We evaluated $S$ with $n = \{1, 2, 3, 4, 5\}$ nodes and also the $S$ composed of all $N$ nodes, which refer to as "all", as a control. We exhaustively examined all possible $S$ with $n = 1$ or $n = 2$ and uniformly randomly sampled 5000 sets of $S$ for each $n \in \{3, 4, 5\}$ due to a large number of combinations. In Supplementary Note 4, we also provide a stopping

criterion and a numerical demonstration when one wants to explore $S$ with $n > 5$ nodes for better solutions (i.e., $S$ with larger $d$ values).

We show the results of one set of simulations on the Barabási-Albert (BA) model network with 50 nodes and average degree $\langle k \rangle = 3.88$ in Fig. 6. We gradually increased $u$ as the bifurcation parameter. Figure 6a shows the relationship between the Kendall's $\tau$ and $d$ when the early warning signal is the variance of a single $x_i$. Each symbol corresponds to a node. We find that the node maximizing $d$, indicated by the intersection of the two dashed lines, is the third best performer in terms of $\tau$. At each value of $n \in \{2, 3, 4, 5\}$, the maximizer of $d$ is not the top performer in terms of $\tau$ but is a reasonably good performer (see Fig. 6b–e). For example, the pair of nodes attaining the largest $d$, corresponding to the symbol at the intersection of the two dashed lines in Fig. 6b, provides the eighth best early warning signal among all the $N(N-1)/2 = 1225$ pairs of nodes. At each value of $n \in \{1, \ldots, 5\}$, we find that $\tau$ and $d$ are positively correlated; the Pearson correlation coefficient between $\tau$ and $d$ is equal to 0.42, 0.46, 0.45, 0.44, and 0.43 for $n = 1, 2, 3, 4$, and 5, respectively. It should also be noted that both $d$ and $\tau$ generally increase as $n$ increases. When all nodes are used, $d$ and $\tau$ are the largest (see Fig. 6f). Therefore, in the present case, using all the nodes is a better choice than using particular combinations of $n \leq 5$ nodes. However, a reasonably large $\tau$ is attained only using $n \leq 5$ nodes if we choose appropriate node sets based on $d$. One can find $S$ that maximizes $d$ only by observing the covariance matrix at two bifurcation parameter values.

We quantified the performance of the optimized node set, i.e., the node set that maximizes $d$ at each value of $n$, using $p_1$ and $p_2$. For a given $n$, we define $p_1$ as twice the fraction of node sets whose $\tau$ is larger than that for the optimized node set. We note that $0 \leq p_1 \leq 2$. We define $p_2$ as $(\tau_{\max} - \tau^*)/(\tau_{\max} - \langle \tau \rangle)$, where $\langle \tau \rangle$ is the average of $\tau$ over all the node sets with $n$ nodes examined, $\tau_{\max}$ is the largest (i.e., best) $\tau$ value among the node sets with $n$ nodes examined, and $\tau^*$ is the $\tau$ value realized by the maximizer of $d$ with $n$ nodes. Although we have assumed that a larger positive $\tau$ is better, the definition of $p_1$ and $p_2$ are similar when the nodes transit from their upper to lower states through tipping points and therefore a larger negative $\tau$ implies a better performance of an early warning signal (Supplementary Note 5). If the maximizer of $d$ is the best node set, realizing the largest $\tau$, then both $p_1$ and $p_2$ are equal to 0 and the smallest. Smaller $p_1$ and $p_2$ values are better. If half the node sets are better than the maximizer of $d$, meaning that the maximizer of $d$ is no better than a uniformly random pick, then we obtain $p_1 = 1$. Similarly, $p_2 = 1$ is equivalent to $\tau^* = \langle \tau \rangle$, implying that the maximizer of $d$ is no better than a random pick. If the maximizer of $d$ is the worst performer, we obtain $p_1 = 2$ and a value of $p_2$ larger than 1.

We show in Fig. 7a the $p_1$ and $p_2$ values for the double-well dynamics on the BA network with $n \in \{1, \ldots, 5\}$. Each small symbol represents the $p_1$ and $p_2$ values for a single series of simulations in which we gradually increased the value of $u$ towards the first transition. Figure 6 shows the relationship between $\tau$ and $d$ in one of the 50 series of simulations whose results are shown in Fig. 7a. The larger symbols in Fig. 7 indicate the average over the 50 series of simulations. We find that the optimizer of $d$ is not a top but reasonably good performer because the $p_1$ and $p_2$ values are substantially smaller than 1 in most runs. The performance of the maximizer of $d$ relative to that of uniformly randomly selected node sets with the same number of nodes, $n$, degrades as $n$ increases.

**Robustness of node set optimization under different heterogeneity scenarios, networks, dynamical systems, and other factors.** To examine the generality of the effectiveness of the node set optimization, we ran simulations, assessed the performance of the early warning signals in terms of the Kendall's $\tau$, and calculated $p_1$ and $p_2$ for the following variations. First, we have assumed that all nodes are homogeneous except with regard to the position in the network. However, as we examined with Fig. 3a and b, different nodes may have different

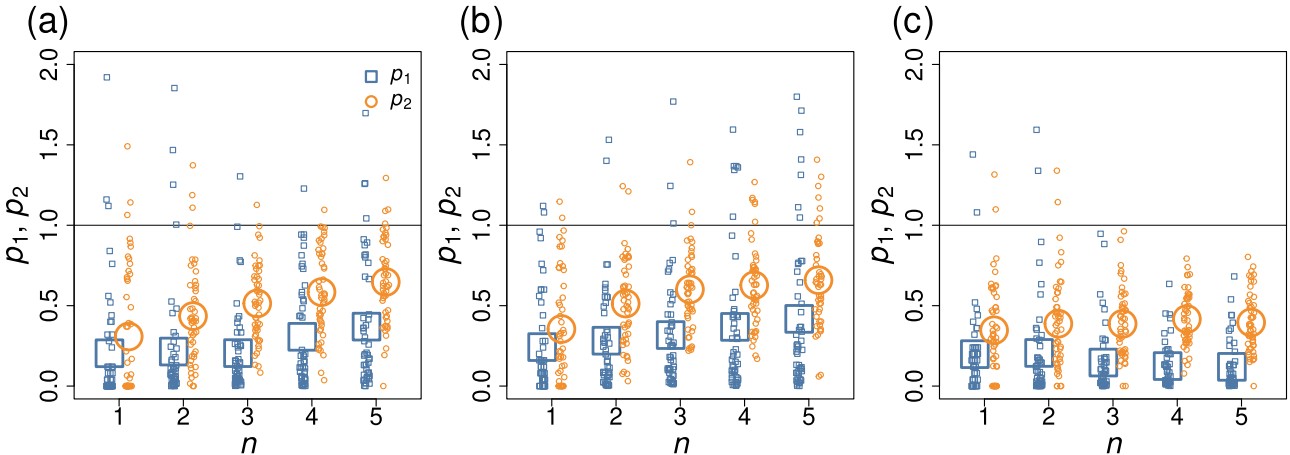

**Fig. 6 | Relationships between the Kendall's $\tau$ and $d$ before the first major transition in the coupled double-well dynamics on a BA network with $N = 50$ nodes.** We gradually increased $u$. We considered node sets $S$ with $n \in \{1, 2, 3, 4, 5, N\}$ nodes. **a** $n = 1$. **b** $n = 2$. **c** $n = 3$. **d** $n = 4$. **e** $n = 5$. **f** $n = N$. The $\tau$ and $d$ values for the node set maximizing $d$ for each $n$ value are highlighted by the dashed lines. The cross represents "Large SD", i.e., the node set comprised of the $n$ nodes with the largest sample standard deviation of $x_i(t)$.

**Fig. 7 | Performance of the optimized node set in anticipating the tipping point in the double-well dynamics on the BA model network with $N = 50$ nodes.** **a** Homogeneous stress and homogeneous noise. **b** Heterogeneous stress and homogeneous noise. **c** Heterogeneous stress and heterogeneous noise. The smaller symbols show the $p_1$ and $p_2$ values for the individual series of simulations. The larger symbols show the average over 50 series of simulations.

propensity to tip, yielding a multistage transition for the entire system. To examine this case, we set $u_i = u + \Delta u_i$ in Eq. (26), where $\Delta u_i$ independently obeys the uniform density on $[-0.25, 0.25]$. We continue to use $u$ as the bifurcation parameter. A node with large $u_i$ tends to transit from its lower state to the upper state earlier as $u$ gradually increases.

Note that a heterogeneous distribution of $u_i$ makes it more difficult to find a good node set $S$ only from the information on the network structure. We show the $p_1$ and $p_2$ values when $u_i$ is heterogeneous and $\sigma_i$ is homogeneous in Fig. 7b. The results are qualitatively the same as those for the case in which both $u_i$ and $\sigma_i$ are homogeneous (see

Fig. 7a). Separately, we also consider the case in which both the node stress $u_i$ and the strength of the dynamical noise $\sigma_i$ are heterogeneous. We set $\sigma_i = \sigma + \Delta\sigma_i$, where $\sigma = 0.05$, and $\Delta\sigma_i$ independently obeys the uniform density on $[-0.9\sigma, 0.9\sigma]$. We show $p_1$ and $p_2$ for this case in Fig. 7c. The results are qualitatively the same as those when both $u_i$ and $\sigma_i$ are homogeneous (Fig. 7a) and when only $u_i$ is heterogeneous (Fig. 7b).

Second, we have considered six networks including the BA model, three of which are model networks and the other three are empirical networks (see Methods for the networks). Third, we have considered four dynamical system models on networks including the coupled double-well dynamics; the other three dynamics are a model of mutualistic interaction dynamics among species[46], a gene regulatory system governed by the Michaelis-Menten equation[46], and the deterministic approximation of the susceptible-infectious-susceptible (SIS) model on networks[47]. These four models of dynamics are diverse in the sense that the SIS model shows a transcritical bifurcation to mark the onset of epidemic spreading, while the other three models show saddle-node bifurcations. In addition, we started the mutualistic interaction and gene regulatory dynamics from the upper state for each node and gradually decreased the bifurcation parameter value, which is the opposite to the case of the double-well and SIS dynamics. This is because the loss of resilience such as species loss in mutualistic dynamics and cell death in gene regulatory dynamics is of a practical concern for these dynamical systems[46]. Fourth, we also varied $D$ instead of $u$ as bifurcation parameter until the first bifurcation occurs. For the SIS model, we only considered an equivalent to $D$, which is the infection rate parameter, but not $u$ because introducing and varying $u$ is unrealistic for the SIS model; see Methods for the discussion.

We show the $p_1$ and $p_2$ values for $n \in \{1, ..., 5\}$, the three different scenarios for $u_i$ and $\sigma_i$ (i.e., constant across all nodes or heterogeneous), two networks, four different dynamical systems, and two different bifurcation parameters (i.e., $u$ or $D$) in Fig. 8. Each figure panel contains the average of $p_1$ and $p_2$ over 50 series of simulations for the three scenarios of heterogeneity (i.e., both $u_i$ and $\sigma_i$ are homogeneous, only $u_i$ is heterogeneous, and both of them are heterogeneous) for one network and one dynamical system. Note that Fig. 8a is for the double-well dynamics on the BA network, corresponding to Figs. 6 and 7. We find that the node set based on the largest $d$ value performs reasonably well for the two networks (see Fig. 8a–g for the BA network and Fig. 8h–n for the Chesapeake Bay carbon flow network) and for all the four models of dynamics. The node set maximizing $d$ tends to work better relative to uniformly randomly picked node sets, yielding smaller $p_1$ and $p_2$ values, when $u_i$ and $\sigma_i$ are both heterogeneous across the nodes (shown by the dotted lines) than when either $u_i$ or $\sigma_i$ is homogeneous (shown by the solid and dashed lines). This is presumably because the proposed method actively discards nodes with high intrinsic noise by comparing the distribution of the signal variance at two values of the bifurcation parameter. The node set maximizing $d$ tends to work better relative to uniformly random node sets with the same number of nodes, $n$, when $n$ is smaller. However, even at $n = 5$, the $p_1$ and $p_2$ values are at most approximately 0.7 and much smaller in a majority of cases. The results are qualitatively the same for the other four networks, with a caveat that the maximizer of $d$ performs poorly for the gene regulatory dynamics model on some networks (see Supplementary Note 6).

We also carried out the following robustness tests.

Our choice of the two bifurcation parameter values at which we sample the covariance matrix and then calculate $d$ (see Methods for the definition of the two bifurcation parameter values) has been arbitrary. We used one bifurcation parameter value close to the start of each series of simulation and the other value close to the first tipping point, regardless of the type of bifurcation. This is because, with this choice, the difference between $\mu_1$ and $\mu_2$, which is the numerator of $d$ (see Eq. (15)), is larger than when the two bifurcation parameter values

are closer. Then, the contrast of $d$ for different choices of $S$ may be larger, potentially helping to single out the $S$ that maximizes $d$. To test the robustness of our results with respect to the choice of the two bifurcation parameter values, we measured $p_1$ and $p_2$ for various pairs of the bifurcation parameter value, which are different in terms of the separation between the two values and the closeness to the first tipping point. For the three scenarios for $u_i$ and $\sigma_i$ (i.e., constant across all nodes or heterogeneous), six networks, four dynamical systems, and two bifurcation parameters (i.e., $u$ or $D$), we have found that the result that $p_1$ and $p_2$ are small in most cases is robust. Specifically, $p_1$ and $p_2$ are considerably smaller than 1 when they are so for the original pairs of the bifurcation parameter value, the two bifurcation parameter values are not too close to each other, and the larger bifurcation parameter value is reasonably close to the first tipping point. (See Supplementary Note 2 for the numerical results.)

Not all the regime shifts accompany critical slowing down. Early warning signals based on critical slowing down, including $\hat{V}_S$ for any node set $S$, should be insensitive to such types of regime shifts[1,11,14,48]. To verify that this is the case, we investigated coupled double-well dynamics on the BA network experiencing a state transition from the lower to the upper state of $x_i(t)$ driven by either dynamical noise or impulse input given to $x_i(t)$'s. Dynamical noise and impulse input are two major scenarios in which regime shifts occur without critical slowing down[11,14]. It should be noted that, in both scenarios, all the system parameter values remain the same throughout a simulation. As expected, $\hat{V}_S$ is found to be insensitive to impending regime shifts until they are about to occur, and therefore, $\tau$ is close to 0. These results hold true for any choice of $S$, and the $S$ maximizing $d$ does not yield a particularly large $\tau$ value. See Supplementary Note 7 for details.

In contrast to the last case, critical slowing down can occur even when there is no sudden regime shift. Representative examples of this phenomenon are transcritical and Hopf bifurcations[14,49]. The gene regulatory and SIS models that we have employed show transcritical bifurcations, at least in well-mixed populations. We showed that $\hat{V}_S$ increases just before the bifurcation, which is captured by large $\tau$ values, in the gene regulatory and SIS models (see Supplementary Note 8 for numerical evidence). Therefore, consistent with the previous studies[14,49], we are not claiming that $\hat{V}_S$ specifically increases when a bifurcation accompanying a large regime shift, which is typically a saddle-node bifurcation, is being approached.

Last, we derived theory for the variance of $x_i(t)$ and its average over nodes for mathematical convenience. However, the standard deviation of $x_i(t)$ rather than the variance of $x_i(t)$ is also a common early warning signal for anticipating tipping points[13,50,51]. We verified that the results on the advantage of the maximizer of $d$ remained almost the same when we used the average of the sample standard deviation of $x_i(t)$ over $i \in S$ as the early warning signal (see Supplementary Note 9).

**Comparison with other node selection methods.** A simple heuristic to select a node set for constructing early warning signals by averaging is to use the $n$ nodes with the largest standard deviation of $x_i$[29], which we refer to as "Large SD". While there are also other strategies to select node sets, here we compare ours with Large SD because the measurement of the standard deviation of $x_i$ does not require the information about the network structure, and therefore it is fair to compare Large SD and the present method. The crosses in Fig. 6 indicate the Kendall's $\tau$ value for the double-well dynamics on the BA network when we use the $n$ nodes with the largest standard deviation to calculate the early warning signal. Large SD detects the best single node, which the maximization of $d$ does not (see Fig. 6a), Large SD also outperforms the maximizer of $d$, yielding larger $\tau$ values, when $n \in \{2, 3, 4, 5\}$ (see Fig. 6b–e), whereas the advantage of Large SD is marginal for $n = 5$. Therefore, when $u_i$ and $\sigma_i$ are homogeneous, Large SD beats the maximizer of $d$, at least for this example. However, the maximizer of $d$ usually behaves much better than Large SD when $\sigma_i$ is heterogeneous.

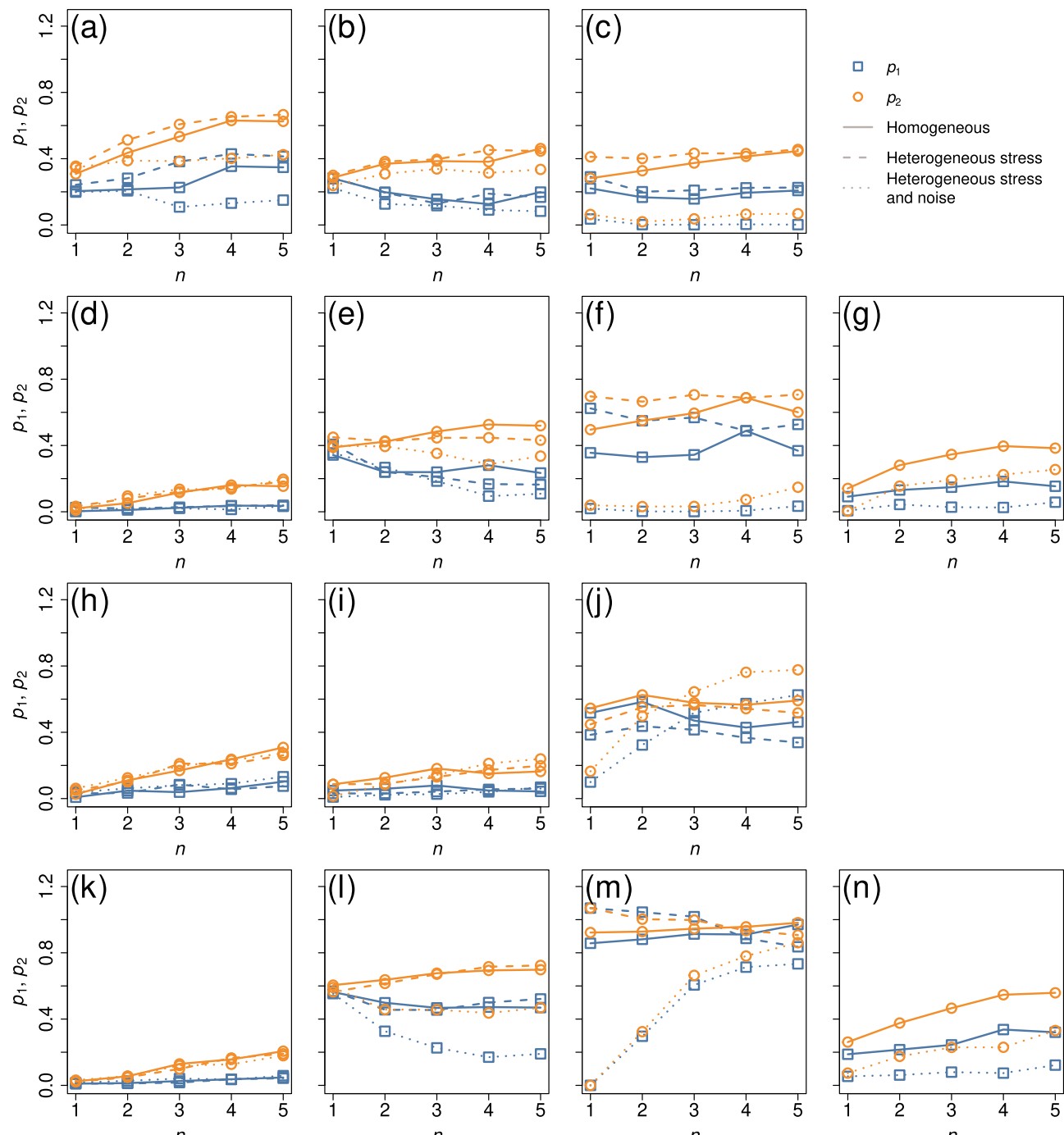

**Fig. 8 | Performance of the node set maximizing $d$ on the BA and Chesapeake Bay networks.** The squares and circles represent $p_1$ and $p_2$, respectively, for the given $n$, dynamics, network, and condition (i.e., whether $u_i$ or $\sigma_i$ is homogeneously or heterogeneously distributed) averaged over 50 series of simulations. **a**–**g** BA network. **h**–**n** Chesapeake Bay carbon flow network. The panels on the leftmost column correspond to the double-well dynamics, and the second to the fourth columns to the mutualistic interaction, gene regulatory, and SIS dynamics, respectively. The combination of the dynamics model and bifurcation parameter is as follows. (**a**, **h**): double-well, $u$. (**b**, **i**): mutualistic interaction, $u$. (**c**, **j**): gene regulatory, $u$. (**d**, **k**): double-well, $D$. (**e**, **l**): mutualistic interaction, $D$. (**f**, **m**): gene regulatory, $D$. (**g**, **n**): SIS, $\lambda$.

Figure 9 shows an example. In this figure, we used the double-well dynamics on the BA network, as we did for Fig. 6, and assumed that both $u_i$ and $\sigma_i$ are heterogeneous. The figure indicates that the maximizer of $d$ substantially outperforms Large SD except when $n = 4$ and $n = 5$, for which the maximizer of $d$ only slightly outperforms Large SD. The compromised performance of Large SD in this scenario is because Large SD tends to select nodes with large $\sigma_i$ although a large $\sigma_i$ value simply reflects that the $i$th node is inherently noisier than others.

To compare between the maximizer of $d$ and Large SD more systematically, we compared the Kendall's $\tau$ values attained by these two strategies for the six networks, the four dynamics models, and whether the stress or dynamical noise is homogeneous or heterogeneous across the nodes. As we show in Supplementary Note 8, we confirmed that the maximizer of $d$ outperforms Large SD in a majority of cases when both the stress, $u_i$, and the noise strength, $\sigma_i$, are node-dependent. The maximizer of $d$ outperforms Large SD even when $\sigma_i$ is node-independent for the mutualistic interaction dynamics and $D$

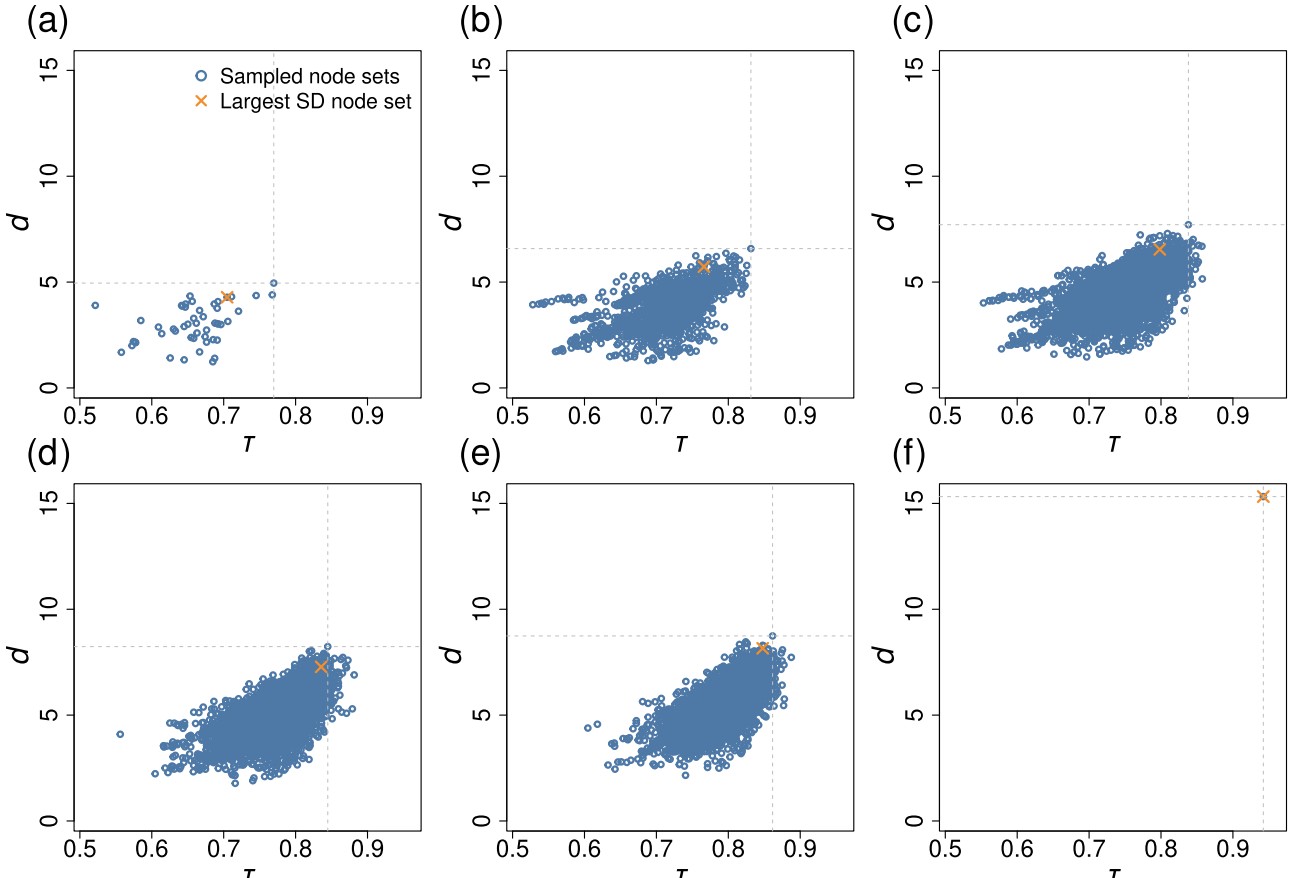

**Fig. 9 | Relationships between the Kendall's $\tau$ and $d$ in the coupled double-well dynamics on the BA network with $N = 50$ nodes under heterogeneous node stress and heterogeneous dynamical noise.** The network is the same as the one used in Fig. 6. We considered node sets $S$ with $n \in \{1, 2, 3, 4, 5, N\}$ nodes. **a** $n = 1$. **b** $n = 2$. **c** $n = 3$. **d** $n = 4$. **e** $n = 5$. **f** $n = N$. The $\tau$ and $d$ values for the node set maximizing $d$ are highlighted by the dashed lines. The cross represents "Large SD", i.e., the node set comprised of the $n$ nodes with the largest sample standard deviation of $x_i(t)$.

being used as the bifurcation parameter. In the other cases, the maximizer of $d$ is either on par with or slightly inferior to Large SD. However, there is no case in which Large SD substantially outperforms the maximizer of $d$ (i.e., Large SD outperforms the maximizer of $d$ in terms of $\tau$ by at most $\approx 0.078$), whereas the maximizer of $d$ often substantially outperforms Large SD.

Furthermore, we compared the maximizer of $d$ with another heuristic algorithm with which one selects the $n$ nodes that receive the highest total input (or lowest total input, depending on the direction of the bifurcation being considered) from the other nodes near the bifurcation point[29], which we refer to as "High/Low Input". The results of comparison between the maximizer of $d$ and High/Low Input were similar to the case of the comparison between the maximizer of $d$ and Large LD (see Supplementary Note 10 for the methods and results). In other words, the maximizer of $d$ tends to be better than the High/Low Input algorithm, in particular when the nodes' dynamics are assumed to be heterogeneous. We note that High/Low Input requires the adjacency matrix of the network, i.e., complete information on the network structure, which neither the maximizer of $d$ nor Large SD requires.

## Discussion

Based on theory of OU processes, we proposed an objective function $d$ for identifying sets of nodes, $S$, that are expected to reliably alert impending tipping events when we combine the early warning signals from $S$ by averaging. While we focused on anticipating the first bifurcation in the entire network, the proposed method is equally applicable to anticipating the second and later transitions in the case of multistage transitions in which different nodes experience regime

shifts at different timings[11,25–29]. We numerically demonstrated the proposed method with various dynamics models, networks, and system heterogeneity scenarios to confirm its good performances. Application to domain-specific problems such as in ecology, climate dynamics, and disease progression is saved for future work. In these and other applications, further complexity of systems in question in addition to the network structure and nonlinearity that we have neglected, such as the mixture of positive and negative interactions[18] and nonequilibrium dynamics[32], may require alternations of our method, posing further research questions.

Nonetheless, our method is readily applicable to empirical data because it does not require the information about the network structure or the dynamical model equations. It only requires the covariance matrix among the observables measured at two values of the bifurcation parameter, or two different states of the networked system, to determine an optimal sentinel node set, $S$. In ecological[52,53] and psychopathological[54] applications of early warning signals, it is customary to calculate fluctuation-based early warning signals such as the variance or covariance from multivariate time series data with sliding time windows, thus obtain time series of early warning signals, and use them to give alerts for impending tipping points. To apply our methods to empirical multivariate time series data, once one has determined $S$ as the maximizer of $d$, one only needs to collect signals from the nodes in $S$ in each of the sliding time windows, calculate the early warning signal (i.e., $\hat{V}_S$), and monitor it.

The proposed method tended to outperform other heuristics when the amount of dynamical noise depended on nodes. The assumption of heterogeneous noise, also made in a previous analytical

study[24], is probably realistic because the nodes constituting a complex dynamical system correspond to different species or geographical patches in ecosystems, different genes or symptoms in complex systems of diseases, different firms or countries in a financial network, and so on. Although it is not easy to measure dynamical noise for each node in isolation in empirical complex systems, there is no a priori reason to assume that different nodes are subject to the same amount of intrinsic noise. As we have shown, heterogeneous noise across nodes generally confuses early warning signals estimated from observed data because, in heterogeneous noise cases, a larger variance of an $x_i$ does not imply that $x_i$ provides a better early warning signal. We overcome this problem by quantifying fluctuations of the early warning signal, not just fluctuations of $x_i$, and measuring it at two bifurcation parameter values. We propose that we should observe the system at two sufficiently distant bifurcation parameter values (practically, two distant times) for identifying good early warning signals. The early warning signals at nodes whose fluctuation is large due to large intrinsic noise do not substantially grow between the two bifurcation parameter values. In contrast, the early warning signals at nodes closely approaching their tipping point substantially grow between the two bifurcation parameter values. An alternative strategy is to use lagged autocorrelation as early warning signal because nodes with large intrinsic noise may produce small autocorrelation. Autocorrelation of multivariate OU processes is analytically tractable, whereas it is more complicated than the variance and covariance[37]. Analysis of autocorrelation and its average over nodes as early warning signals with the present theoretical framework warrants future work.

In numerical simulations, we examined the size of the node set $n \in \{1, ..., 5\}$. For $n = 1$ and $2$, we inspected all the node sets for their performance and $d$ values. In contrast, we randomly sampled node sets for $n \geq 3$ due to the combinatorial explosion. If the number of nodes, $N$, is less than approximately 20, it may be feasible to exhaustively investigate all the node sets across all values of $n$ to find the exact maximizer of $d$. Note that the calculation of a single $d$ only involves the square and summation of the entry of the $N \times N$ covariance matrix and therefore is not costly. When $N$ is larger, we need to find node sets that only approximately maximize $d$. This combinatorial problem has scarcely been discussed in early warning signal research community. It is because prior studies highlighting that different nodes can emit early warning signals of different quality either used systems with a small number of nodes, typically with $N \leq 5^{19,21-24}$, or linearly ranked the $N$ nodes[20,27,29], thus implicitly abandoning the potential benefit of wisely combining different nodes. However, real-world complex systems whose tipping events are of practical interest are more often than not larger complex networks[17,18]. Although we provided a stopping criterion for exploring different $n$ values and gave a demonstration (see Supplementary Note 4), further deploying heuristics for combinatorial optimization to approximate maximization of $d$ for reasonably large $n$ and $N$ is left for future work.

We used the sample covariance matrix as the estimator of the true covariance matrix for simplicity. In fact, although the sample covariance matrix is an unbiased estimator in the limit of $L \rightarrow \infty$, it is an unreliable estimator when $L$ is small relative to $N$, and there are better choices such as different sparse estimators and covariance shrinkage methods[45]. In the present study, the sample covariance matrix is not problematic because we used a relatively small $N$ and large $L$. However, in practical situations, we may only have a small number of samples, in which case one should consider a different covariance estimator. Furthermore, we confined ourselves to linear combinations of early warning signals measured at single nodes. However, a natural extension is to exploit covariance of $x_i$ and $x_j$, where $i \neq j$, to construct early warning signals. Examples of such early warning signals include the leading eigenvalue of the covariance matrix[21,55,56] and Moran's $I$[57-59]. It should be noted that we used the cross covariance, $C_{ij}^{(1)}$ and $C_{ij}^{(2)}$, where $i \neq j$, only to evaluate the uncertainty of our early warning signals, not

to construct early warning signals. Early warning signals for heterogeneous networks when the number of samples is limited are a challenging open question.

## Methods

### Dynamical system models

We used the following four types of dynamics on networks with dynamical noise.

A coupled double-well model on networks with dynamical noise is given by Eq. (26). We set $(r_1, r_2, r_3) = (1, 3, 5)$. In the presence of coupling (i.e., $D > 0$), the lower and the upper state of each node is around $x_i = r_1$ and $x_i = r_3$, respectively. Here we succinctly regard that the nodes with $x_i < r_2$ and $x_i \geq r_2$ are in the lower and the upper state, respectively. We set $u_i = u + \Delta u_i \ \forall \ i$. If $u$ is the bifurcation parameter, we initially set $u = 0$ and $D = 0.05$. If $D$ is the bifurcation parameter, we initially set $u = 0$ and $D = 0$. In the case of homogeneous stress, we set $\Delta u_i = 0 \ \forall \ i$. In the case of heterogeneous stress, we draw each $\Delta u_i$ independently from the uniform density on $[-0.25, 0.25]$ for this and the following three models of dynamics. We set $\sigma_i = \sigma + \Delta \sigma_i$, where $\sigma = 0.05$ for this model. We set $\Delta \sigma_i = 0 \ \forall \ i$ in the homogeneous noise case and draw $\Delta \sigma_i$ from the uniform density on $[-0.9\sigma, 0.9\sigma]$ in the heterogeneous noise case, including for the other three dynamical system models described in the remainder of this section.

The mutualistic interaction dynamics among species is given by

$$\mathrm{d}x_i = \left[ B_i + x_i \left( 1 - \frac{x_i}{K_i} \right) \left( \frac{x_i}{C_i} - 1 \right) + D \sum_{j=1}^{N} w_{ij} \frac{x_i x_j}{\tilde{D}_i + E_i x_i + H_j x_j} \right] \mathrm{d}t + \sigma_i \mathrm{d} W_i,$$

(27)

where $x_i$ represents the abundance of the $i$th species, and $B_i, C_i, \tilde{D}_i, E_i, H_i$, and $K_i$ (with $i \in \{1, ..., N\}$) are constants[46]. Constant $B_i$ represents the migration rate of the $i$th species from outside the entire ecosystem. The second term on the right-hand side of Eq. (27) represents the logistic growth with carrying capacity $K_i$ and Allee constant $C_i$. The third term represents the mutualistic effect of the $j$th species on the $i$th species under the assumption that $w_{ij} \geq 0$. This term is bounded by definition for preventing $x_i$ to exceed $K_i$ to break down the logistic growth nature of the second term. We set $B_i = 0.1 + u + \Delta u_i, C_i = 1, \tilde{D}_i = 5, E_i = 0.9, H_i = 0.1$, and $K_i = 5 \ \forall \ i \in \{1, ..., N\}$, following ref. 46. Regardless of whether $u$ or $D$ is the bifurcation parameter, we initially set $u = 0$ and $D = 1$. We confirmed that all $x_i$ were in their upper state (i.e., $\gg 0$) at equilibrium when $(u, D) = (0, 1)$. We set $\sigma_i = \sigma + \Delta \sigma_i$ with $\sigma = 0.25$.

The gene regulatory dynamics is given by

$$\mathrm{d}x_i = \left( -Bx_i^f + D \sum_{j=1}^{N} w_{ij} \frac{x_j^h}{x_j^h + 1} + u_i \right) \mathrm{d}t + \sigma_i \mathrm{d}W_i,$$

(28)

where $x_i$ represents the expression level of the $i$th gene[46]. We set $B = 1, f = 1$, and $h = 2$ by following ref. 46, and also $u_i = u + \Delta u_i$. Regardless of whether $u$ or $D$ is the bifurcation parameter, we initially set $u = 0$ and $D = 1$. Then, we obtain $x_i > 0.1 \ \forall \ i$ at equilibrium if the initial values of $x_i \ \forall \ i$ are sufficiently large, which we assume. We set $\sigma_i = \sigma + \Delta \sigma_i$, where $\sigma = 5 \times 10^{-6}$.

The SIS dynamics on networks with dynamical noise is given by

$$\mathrm{d}x_i = \left[ \lambda \sum_{j=1}^{N} w_{ij}(1 - x_i)x_j - \mu x_i \right] \mathrm{d}t + \sigma_i \mathrm{d} W_i,$$

(29)

where $x_i$ represents the probability that the $i$th node is infectious, $\lambda$ is the infection rate (specifically, the rate at which the $i$th node is infected by an infectious neighbor) and is equivalent to $D$ in the coupled double-well, mutualistic interaction, and gene regulatory dynamics models, and $\mu$ is the recovery rate. The first term on the right-hand side of

Eq. (29) represents that the $j$th node infects the $i$th node. The second term represents the recovery of the $i$th node. We set $\mu = 1$ without loss of generality; multiplying $\lambda$, $\mu$, and $\sigma_i$ by the same positive constant is equivalent to changing the time scale and not to change these parameter values. For this model, we only use $\lambda$ as the bifurcation parameter. This is because adding a term $u_i dt$ on the right-hand side of Eq. (29) would imply that infectious individuals can spontaneously appear in the population in the absence of any infected host, which is unrealistic. For the same reason, we do not have a concept of heterogeneous node stress in this model. We initially set $\lambda = 0$. In the absence of dynamical noise, it holds true that $0 \le x_i < 1 \ \forall i$ at equilibrium. We set $\sigma_i = \sigma + \Delta\sigma_i$, where $\sigma = 5 \times 10^{-4}$.

### Networks
We used the following three model networks and three empirical networks in our numerical simulations. All networks are undirected and unweighted by construction, or coerced to be so.

The first network was generated by the Erdős-Rényi random graph with 50 nodes and exactly 125 edges. We connected a uniformly randomly selected pair of nodes that had not yet been adjacent to each other, one by one, until we had 125 edges. It happened that the generated network was connected. By construction, the average degree $\langle k \rangle = 5$.

The second network is a network generated by the BA model with $N = 50$ nodes. We set the number of edges per each additional node to $m = 2$. The initial condition was the complete graph with 3 nodes (i.e., triangle). In the limit $N \to \infty$, the model produces a degree distribution with a power-law tail, i.e., $p(k) \propto k^{-3}$, where $k$ is the degree, and $p(k)$ is the probability that the degree is equal to $k$. The resulting network was connected and had 97 edges, yielding $\langle k \rangle = 3.88$.

The third network is the largest connected component of the node fitness model proposed by refs. [60–62]. This model produces networks with heterogeneous degree distributions as follows. We start with an empty network with $N = 50$ nodes and assign to each node $i$ a fitness score $f_i = (i + i_0 - 1)^{-\alpha}$, where $\alpha$ is a parameter that controls the heterogeneity of the degree distribution, and $i_0 = N^{1-\frac{1}{\alpha}} [10\sqrt{2}(1 - \alpha)]^{\frac{1}{\alpha}}$ constrains the maximum degree. We set $\alpha = 2$. Then, we independently connect each pair of nodes $(i, j)$ by an edge with probability $f_i f_j / (\sum_{\ell=1}^N f_\ell)^2$. We took the largest connected component of the resulting network, which included $N = 49$ nodes and 125 edges, yielding $\langle k \rangle = 5.1$.

The fourth network is a record of carbon flows in the Chesapeake Bay marine ecosystem[63]. The original carbon flow data is directed and forms a connected network. We used the undirected, unweighted version of the network stored by the Koblenz Network Collection[64]. This network has 39 nodes, 170 edges, and $\langle k \rangle = 8.72$.

The fifth network is the food web of a freshwater stream collected in southern New Zealand[65]. Two species are connected if there was evidence of one consuming the other. The original data is a directed and unweighted network with 49 nodes and 110 edges[66]. We coerced the network to be undirected and retained the largest connected component, resulting in a network with 48 nodes, 110 edges, and $\langle k \rangle = 4.58$

The sixth network is a social network of wild dolphins observed in Doubtful Sound, New Zealand[67]. The network is connected, undirected, and unweighted, with 62 nodes, 159 edges, and $\langle k \rangle = 5.13$.

### Calculation and performance assessment of early warning signals
We conducted 50 independent series of simulations for each combination of dynamics model, network, and whether both node stress (i.e., $u_i$) and noise strength (i.e., $\sigma_i$) were homogeneous, only $u_i$ was heterogeneous, or both $u_i$ and $\sigma_i$ were heterogeneous. Each series consisted of simulations at linearly increasing values of a bifurcation

parameter in the case of the double-well or SIS dynamics, and linearly decreasing values of a bifurcation parameter in the case of the mutualistic interaction or gene regulatory dynamics. When we moved to a next value of the bifurcation parameter, we increased it by $\Delta u = 0.025$ or $\Delta D = 0.0025$ for the double-well dynamics, decreased it by $\Delta u = 0.1$ or $\Delta D = 0.01$ for the mutualistic interaction dynamics, decreased it by $\Delta u = 0.01$ or $\Delta D = 0.01$ for the gene regulatory dynamics, and increased it by $\Delta\lambda = 0.0025$ for the SIS dynamics.

Each simulation given the dynamics model began from the same value of $x_i \ \forall i$ (double-well: $x_i = 1$, mutualistic interaction: $x_i = 5$, gene regulatory: $x_i = 5$, SIS: $x_i = 0.001$) regardless of the bifurcation parameter value. We used the Euler-Maruyama method with $\Delta t = 0.01$ to simulate each dynamics. In the SIS and gene regulatory dynamics, whenever we obtain a negative value of $x_i(t)$ for any $i$ due to dynamical noise, we reset $x_i(t) = 0$. We allowed 100 generic time units (TU) to discard transients, except in the case of the mutualistic dynamics, for which we allowed 10 TU due to a shorter characteristic time scale of the mutualistic dynamics model. After discarding transients, we considered the system at equilibrium and took $L = 100$ evenly spaced samples from each $x_i(t)$, $i \in \{1, 2, ..., N\}$; samples were spaced 1 TU apart, with the exception of the mutualistic dynamics model for which the samples were spaced 0.1 TU apart. We stopped a series of simulations with a gradually changing bifurcation parameter value once any $x_i$ was no longer near its initial state, specifically, once any $x_i$ satisfies the following condition: $x_i \ge 3$ for the double-well dynamics, $x_i < 0.1$ for the mutualistic interaction dynamics, $x_i < 0.1$ for the gene regulatory dynamics, and $x_i \ge 0.1$ for the SIS dynamics.

After all simulations in a given series were complete, we calculated the early warning signals for each node set $S$ with $|S| = n$, based on the $L$ samples taken at each value of the bifurcation parameter. In addition, to calculate $d$, we used Eq. (5) to obtain $\mu_1 = \frac{1}{n} \sum_{i=1}^n C_{ii}^{(1)}$ and $\mu_2 = \frac{1}{n} \sum_{i=1}^n C_{ii}^{(2)}$, and Eq. (6) to obtain $\mathrm{var}_1 = \frac{2}{n^2(L-1)} \sum_{i=1}^n \sum_{j=1}^n (C_{ij}^{(1)})^2$ and $\mathrm{var}_2 = \frac{2}{n^2(L-1)} \sum_{i=1}^n \sum_{j=1}^n (C_{ij}^{(2)})^2$. Then, we used Eq. (15) to calculate $d$ for the node set $S$. Note that this computation is easy once we have calculated $C^{(1)}$ and $C^{(2)}$. We selected the two values of the bifurcation parameter at which we calculated $C^{(1)}$ and $C^{(2)}$ as follows. Suppose that the bifurcation parameter is $u$ and that we have simulated the dynamics at $u = \bar{u}_k$, $k \in \{1, 2, ..., \tilde{K}\}$. Thus, the simulation with $\bar{u}_{\tilde{K}}$ was the last simulation in which all $x_i$ remained near their initial state at equilibrium. We selected $k^{(1)} = \mathrm{round}(0.1\tilde{K})$ and $k^{(2)} = \mathrm{round}(0.9\tilde{K})$, where round() denotes rounding to the closest integer. Then, we calculated $C^{(1)}$ and $C^{(2)}$ from the $L$ samples obtained at $u = u^{(1)} \equiv \bar{u}_{k^{(1)}}$ and $u = u^{(2)} \equiv \bar{u}_{k^{(2)}}$, respectively. We followed the same procedure to select the two values of $D$ at which we calculated $C^{(1)}$ and $C^{(2)}$ when the bifurcation parameter was $D$.

In addition to by maximizing $d$, we also determined node set $S$ by the Large SD algorithm[29] for comparison purposes, which proceeds as follows. First, we collected $L$ samples from each $x_i(t)$ at $u = u^{(2)}$ (or $D = D^{(2)}$ when $D$ instead of $u$ was the bifurcation parameter) and calculated its sample standard deviation. Second, we defined $S$ as the set of the $n$ nodes with the largest sample standard deviation of $x_i(t)$. Third, we averaged the sample standard deviation of $x_i(t)$ over $i \in S$ and used it as early warning signal.

We quantify the extent to which a given node set $S$ signals the proximity of tipping by the Kendall's $\tau$ rank correlation between the early warning signal (i.e., the sample variance averaged over the nodes in $S$ unless we state otherwise) and the bifurcation parameter[13]. Because of critical slowing down, the variance of $x_i(t)$ grows large as the dynamical system approaches a tipping point[1]. In our simulations, when the bifurcation parameter linearly increases (i.e., double-well and SIS dynamics), the value of a perfect early warning signal would monotonically increase, resulting in $\tau = 1$. When the bifurcation parameter linearly decreases (i.e., mutualistic interaction and gene regulatory dynamics), a perfect early warning signal yields $\tau = -1$.

**Reporting summary**

Further information on research design is available in the Nature Portfolio Reporting Summary linked to this article.

## Data availability

The numerical data that are generated during the current study and underlie the figures in this article are available on Github at https://github.com/ngmaclaren/mixing-EWS.

## Code availability

The code for generating the results and figures in this article is publicly available on Github at https://github.com/ngmaclaren/mixing-EWS.

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

## Acknowledgements

Na.M. and K.A. acknowledge support from the Japan Science and Technology Agency (JST) Moonshot R&D (under grant no. JPMJMS2021). Na.M. acknowledges support from the National Science Foundation (under grant no. 2204936) and JSPS KAKENHI (under grant nos. JP 21H04595 and 23H03414). K.A. acknowledges support from Japan Agency for Medical Research and Development (AMED) (under grant no. JP23dm0307009), JSPS KAKENHI (under grant no. JP20H05921), and Institute of AI and Beyond at the University of Tokyo.

## Author contributions

Na.M. conceived the research, developed the theory, performed the numerical simulations with two- and three-node networks, and drafted the manuscript. Ne.G.M. performed all the other numerical simulations. Na.M., K.A., and Ne.G.M. discussed the results, and revised and checked the manuscript.

## Competing interests

The authors declare no competing interests.
