## [Peer Review File · Nature Communications]

REVIEWER COMMENTS

Reviewer #1 (Remarks to the Author):

Sudden changes in the stable steady states of complex dynamical systems as a system parameter changes, also called regime shifts, are of great concern and constitute a forefront interdisciplinary research field. A key issue is early warnings. For networked dynamical systems, early warning signals can be obtained from a subset of nodes in the network, but how to determine the optimal subset of nodes to extract early warning signals had been unclear. In this paper, the authors developed a mathematical theory based on linear stochastic differential equations to address this important question. The use of linear stochastic system is justified as the system is assumed to be in the vicinity of some stable steady state. The authors focused on regime shift as induced by changes in a bifurcation parameter and proposed an index based on the variances of the nodal state to identify a critical subset of nodes from which early warning signals can be extracted. Because the index is based on a kind of "average variance" of the dynamical state of the nodes, the fluctuations in the early warning signal can be significantly reduced as compared with these in previous works. The authors employed six synthetic and empirical complex networks and four dynamical processes to demonstrate the workings of the proposed method.

The work is well motivated and addresses an important problem of current interest. The manuscript can be improved in a number of ways.

1. (major) It is stated that the average value of the sample covariance over a set of nodes should be used to improve the early warning signal. An index (d) was proposed and maximizing this index would lead to the optimal set of nodes. What is the intuitive picture underlying this method? Are the nodes in the optimal set simply those with the least dynamical noise? In any case, some threshold is needed to determine the set. How is the threshold determined?

Section 2 gives the mathematical reasoning behind the proposed index. What is lacking is an intuitive physical explanation. Especially, why maximizing the index would give the best subset of nodes from which early warning signals can be extracted? For a two-node network, the index defined in Eq. (20) is the distance between two normal distributions, and it is necessary to evaluate this index for node 1, node 2, and the "average" of the two nodes. It was demonstrated that the "average" gives the maximum index value because the averaging process cancels the noises from the two nodes, so it should be used. Could it happen that the noises from the two nodes are not canceled out but are added up? - the superposition of two signals would depend on their phases, leading to constructive or destructive interference. Were the effects of the phases taken into account in calculating the index d ?

The original Sec. 2 is mathematical. Perhaps it should be moved to the Methods section. Instead, a new section giving a physical picture of the selection of the optimal set of nodes is desired, especially considering the broad readership of Nat. Commun.

2. (major) To find the optimal set of nodes, it seems necessary to test all possible subsets of nodes in the network. If the network is large, the computation load can be enormous. In fact, this is an NP-hard problem. All networks tested are relatively small networks - with the number of nodes around 50. Can the authors' method be extended to larger networks?

3. (minor) The first example in Sec. 3, especially Fig. 2, is confusing. It seems that Fig. 2 represents the bifurcation diagram of the system rather than its phase portrait especially considering that the paper focuses on bifurcation induced tipping and samples the system at fixed values of "r," implying that "r" is not a variable.

Regarding the sentence: "We assume that $f(x,r)$ is continuous in terms of x and r for simplicity." It's confusing just by looking at Fig. 2 and if the system shown is discontinuous, then the assumption of continuity in the model should be reevaluated and appropriately discussed.

Moreover, the presence of two stable equilibria in cases where "r" is less than zero ($r < 0$) needs further clarification or justification.

4. (minor) The authors claim that it is necessary to sample the system with at least two values of the bifurcation parameter. What are the real-world implications of this requirement and its impact on the results? How are these two values typically chosen, and is there a preference for selecting values that are distant from each other or proximate to the bifurcation point?

5. (minor) A clear definition of what constitutes a "small" and "large" network should be given. Is it in terms of the computational complexity? Are both the number of nodes and the number edges important for calling a network "large"?

6. (minor) Section 4 needs to be reorganized to improve clarity. For instance, it could be divided into three subsections to present the results of each dynamical system separately

Reviewer #2 (Remarks to the Author):

This paper develops a powerful advance in the detection of early warning signals in networked systems. The authors present an optimization method to determine which nodes of a network are the best ones to monitor in order to extract reliable early warning signals based on phenomena such as critical slowing down. Of especial value is the finding that the method excels despite heterogeneity among nodes in terms of dynamical noise and stress. The reason this is so important is that many important systems are characterized by this heterogeneity among nodes—almost to the point that such systems are sometimes not even recognized as networks. Examples include differing countries or different climate subsystems systems that are strongly linked and subject to tipping point dynamics. Few existing methodologies provide guidance on how/which nodes to monitor with the generality that their analytical framework offers. The effect as measured by p_1 and p_2 seems very strong, and they assessed the method across a range of different networks.

The methodology is sound and the writing and organization are good. The only recommendation I have is that the authors should present some results that will more clearly demonstrate the value of the approach to less specialized readers. For example, they demonstrate the superiority of their approach by generating plots for p_1 and p_2 . This is a perfectly valid and correct method for assessing the approach. However, they should also have some plots that simply show variance versus time for the set S , compared to some random sample of nodes, for the BA model for example. This will better convey the usefulness of the approach to less mathematical audiences. Some of the important but confirmatory

plots could be moved to the SI appendix to make way for such plots that can speak to a general readership.

Reviewer #3 (Remarks to the Author):

In the article “Anticipating regime shifts by mixing early warning signals from different nodes” by N Masuda et al., the authors devise an early warning signal for networks by optimizing the node set using a distance measure of distributions at two different bifurcation parameter values. They also compare their studies with other methods of optimizing node sets for calculating early warning signals of regime shifts. They also validate their approach to combinations of varied networks, stress, and dynamical noise.

I find the work exciting and intriguing to a broad audience, as they deal with developing optimized early warning signals for networks that are of broad applicability to regime shift prediction across diverse domains. The manuscript will add to the existing literature on early warning signals of networks, with a lot of room for expansion within the present framework. However, I have a few comments and suggestions that I strongly feel must be addressed before the manuscript can be accepted for publication.

Below are my comments:

1. In Fig. 3, the sample covariance is distributed over a very small interval of the order 10^{-2} , whereas in Fig. 4(b), the sample covariance is distributed over a larger interval. While this can be attributed to significant noise in the later case, one should remember that in empirical data, where noise cannot be differentiated from data, such an inference could result in erroneous signals. This could be voided by measuring the change in covariance at lag one and using the normalized form of the same as an indicator of regime shift. Testing the change in covariance at higher lags could also improve early warning signals.
2. Secondly, the authors measure the distance d at two different values of the bifurcation parameter (r) for constructing the set S of optimized nodes. The bifurcation theory behind the occurrence of a tipping point makes it evident that changes occur in the state of the system in the vicinity of the tipping. In this study, I understand that the authors select two r values from the bistable region. How do the authors choose the different values of r to measure d in the instances of other bifurcation? I recommend the authors make it more clear whether they choose it by trial or there is some intuition at what values of the bifurcation they calculate the difference in distribution d . Also, I suspect the sensitivity of the d value on the choice of r values. A sensitivity analysis showing the effect of choice of r value on the construction of the set S theoretically or numerically would add to the novelty of the study.
3. Additionally, it is important to note that the entire significance of early warning signals is directed to their utility in anticipating regime shifts in real data. Throwing some light on how one can extend this study to empirical data will add to the generality of the study, which is not apparent in the present form of the manuscript.
4. The study of early warning signals here is only focused on trends obtained from data undergoing a regime shift, and their strength is measured using Kendall's tau. However, it is essential to assess the chance of false negatives (i.e., signals in data not undergoing a regime shift) as well. Clearly, previous

literature has shown evidence of a rise in variance and artificial autocorrelation in data not undergoing regime shift. Therefore, it is necessary to perform an analysis of the early warning signals in data not undergoing regime shifts and investigate the strength of signals in the same. This is absent in the present study.

Here are some important references that the authors might find relevant.

Boettiger C, Hastings A. Early warning signals and the prosecutor's fallacy. *Proceedings of the Royal Society B: Biological Sciences*. 2012 Dec 7;27 (1748):4734-9.

Boettiger C, Ross N, Hastings A. Early warning signals: the charted and uncharted territories. *Theoretical ecology*. 2013 Aug;6:255-64.

5. In Fig. 5, the authors show that increasing the size of N improves the early warning. However, in another instance, the authors mention, "The node set maximizing d tends to work better for smaller node sets (i.e., smaller n)." This may be made clear.

6. I find that a limitation of the algorithm is that there are no stopping criteria in constructing S . I think the authors may find it interesting to derive stopping criteria in the construction of the set S as increasing the size of N improves the signal, and this can lead one to continue calculating until one reaches the size of the network. This may be computationally very expensive and undermines the utility of the same when working with higher dimensional real networks.

Reviewer #1

Sudden changes in the stable steady states of complex dynamical systems as a system parameter changes, also called regime shifts, are of great concern and constitute a forefront interdisciplinary research field. A key issue is early warnings. For networked dynamical systems, early warning signals can be obtained from a subset of nodes in the network, but how to determine the optimal subset of nodes to extract early warning signals had been unclear. In this paper, the authors developed a mathematical theory based on linear stochastic differential equations to address this important question. The use of linear stochastic system is justified as the system is assumed to be in the vicinity of some stable steady state. The authors focused on regime shift as induced by changes in a bifurcation parameter and proposed an index based on the variances of the nodal state to identify a critical subset of nodes from which early warning signals can be extracted. Because the index is based on a kind of "average variance" of the dynamical state of the nodes, the fluctuations in the early warning signal can be significantly reduced as compared with these in previous works. The authors employed six synthetic and empirical complex networks and four dynamical processes to demonstrate the workings of the proposed method.

The work is well motivated and addresses an important problem of current interest. The manuscript can be improved in a number of ways.

We are glad to hear an overall positive evaluation by the reviewer. We amended the manuscript according to the reviewer's valuable comments as follows.

1. (major) It is stated that the average value of the sample covariance over a set of nodes should be used to improve the early warning signal. An index (d) was proposed and maximizing this index would lead to the optimal set of nodes. What is the intuitive picture underlying this method? Are the nodes in the optimal set simply those with the least dynamical noise? In any case, some threshold is needed to determine the set. How is the threshold determined?

Intuitively, the nodes in the optimal set tend to be those with the least dynamical noise, but importantly, it is not the whole story. That is, if x_i s for the nodes with the least dynamical noise are correlated with each other (e.g., if those nodes are adjacent to each other by being directly connected by an edge in the network), averaging the variance of x_i over these nodes does not help to reduce the fluctuation of the early warning signal. We added text to the beginning of section 4 (before entering section 4.1) to explain this intuitive picture as a motivator to numerically analyze the case of networks with larger numbers of nodes, N .

Maximization of d is a combinatorial optimization problem. Therefore, there is no threshold involved. On the other hand, when N is large, we cannot obtain the exact maximizer of d due to the combinatorial explosion. Therefore, we

heuristically determined the optimizer of d already in the previous version of the manuscript, which is described in detail (i.e., “We exhaustively examined all possible S with $n = 1$ or $n = 2$ and uniformly randomly sampled 5000 sets of S for each $n \in \{3, 4, 5\}$ due to a large number of combinations.” on lines 271–272 in the revised manuscript). For further improvement of this combinatorial optimization aspect, please see our response to item “2.” below.

Section 2 gives the mathematical reasoning behind the proposed index. What is lacking is an intuitive physical explanation. Especially, why maximizing the index would give the best subset of nodes from which early warning signals can be extracted? For a two-node network, the index defined in Eq. (20) is the distance between two normal distributions, and it is necessary to evaluate this index for node 1, node 2, and the “average” of the two nodes. It was demonstrated that the “average” gives the maximum index value because the averaging process cancels the noises from the two nodes, so it should be used. Could it happen that the noises from the two nodes are not canceled out but are added up? - the superposition of two signals would depend on their phases, leading to constructive or destructive interference. Were the effects of the phases taken into account in calculating the index d ?

First, some intuitive explanation was already in the previous version of the manuscript, i.e., the text earlier in the paragraph containing Eq. (15) in the revised manuscript. We added more text for intuitive explanation in the same paragraph before and after Eq. (15). Furthermore, we added a paragraph discussing the intuitive reason for our choice of d (i.e., the paragraph starting with “Our choice of” in section 4.3).

Second, for general random variables Z_1 and Z_2 , one obtains

$$\text{std}[Z_1 + Z_2] = \sqrt{(\text{std}[Z_1])^2 + \text{std}[Z_2]^2 + 2\text{cov}(Z_1, Z_2)} \leq \text{std}[Z_1] + \text{std}[Z_2], \quad (1)$$

where std denotes the standard deviation, cov denotes the covariance, and the equality holds true if and if Z_1 and Z_2 are perfectly correlated (such that $\text{cov}(Z_1, Z_2) = \text{std}[Z_1]\text{std}[Z_2]$). Therefore, the noises from the two nodes are indeed added up if we just add the early warning signals at node 1 and that at 2 and if these two signals are perfectly correlated. However, we are considering the “average” of the two nodes, as the reviewer also mentioned. That is, we set $\hat{V}_{\{1,2\}} \equiv (\hat{V}_1 + \hat{V}_2)/2$. For the average, Eq. (1) is adapted to

$$\text{std} \left[\frac{Z_1 + Z_2}{2} \right] \leq \frac{\text{std}[Z_1] + \text{std}[Z_2]}{2} \leq \max(\text{std}[Z_1], \text{std}[Z_2]). \quad (2)$$

Therefore, it does not happen that noises from the two nodes are added up in our framework. The fluctuation (i.e., standard deviation) of $\hat{V}_{\{1,2\}}$ is never greater than that of \hat{V}_1 or \hat{V}_2 .

We are not particularly considering phasic signals, but the argument is the same. For example, if one adds two same sinusoidal curves that are in phase, then its amplitude is doubled. However, if we divide the sum by two, then it is

the same as one original sinusoidal curve. If two sinusoidal curves have a phase difference, the sum of them has an amplitude larger than a single sinusoidal curve but smaller than twice of them. Again, the average means to divide the sum by two, so the averaged sinusoidal curve has an amplitude smaller than a single sinusoidal curve. Although we have omitted noise in this argument, we believe that this argument captures possible consequences in the case of phasic signals.

We opted not to include these arguments in the manuscript because we think that they will rather confuse readers (e.g., our theory and models are not about phasic signals/noises).

The original Sec. 2 is mathematical. Perhaps it should be moved to the Methods section. Instead, a new section giving a physical picture of the selection of the optimal set of nodes is desired, especially considering the broad readership of Nat. Commun.

We moved most of the mathematical derivations in section 2 to section S1. Because the remaining content of section 2 has the description of the class of dynamics models we are considering, the description of our main idea, the definition of \hat{V}_S , and intuitive interpretation of our theoretical results, we opt to keep it in section 2 rather than completely hiding them in the Methods section. We are fully aware of the broad readership of Nature Communications, but many Nat Commun papers do have more than a few equations. With the equations and text remaining in section 2, we believe that it is now a smooth read for broad readers.

Our mathematics-based method selects the optimal set of nodes that does not necessarily agree with a physical picture/interpretation. Therefore, we contend that our node set selection is mathematical/algorithmic. Therefore, we did not opt to transform this section into a new section giving a physical picture of the methods. However, to guide readers, we supplied text to explain the intuition behind our node set selection algorithm; for such additional text, please see our response to “4.” below.

2. (major) To find the optimal set of nodes, it seems necessary to test all possible subsets of nodes in the network. If the network is large, the computation load can be enormous. In fact, this is an NP-hard problem. All networks tested are relatively small networks - with the number of nodes around 50. Can the authors' method be extended to larger networks?

Indeed, it is an NP-hard problem, which is why we heuristically (and therefore only approximately) optimized d . We believe that the following sentence (lines 271–272), which had been in the previous version of manuscript already, clarifies this point and implies that one can run the same heuristic optimization for larger networks:

“We exhaustively examined all possible S with $n = 1$ or $n = 2$ and uniformly randomly sampled 5000 sets of S for each $n \in \{3, 4, 5\}$ due to a large number of combinations.”

To address the problem of the performance (in terms of d) and computational efficiency, we also devised a stopping criterion and carried out a numerical demonstration. We briefly discussed this, including the motivation, in the main text (lines 272–273 and 455–457) and provided the details, including the numerical results, in the new section S4. We also provided code to maximize d with the proposed stopping criterion (please see section S4).

3. (minor) The first example in Sec. 3, especially Fig. 2, is confusing. It seems that Fig. 2 represents the bifurcation diagram of the system rather than its phase portrait especially considering that the paper focuses on bifurcation induced tipping and samples the system at fixed values of “ r ,” implying that “ r ” is not a variable.

Indeed, r is not a variable but a bifurcation parameter. To avoid the confusion, we replaced “phase portrait” by “bifurcation diagram”.

Regarding the sentence: “We assume that $f(x,r)$ is continuous in terms of x and r for simplicity.” It’s confusing just by looking at Fig. 2 and if the system shown is discontinuous, then the assumption of continuity in the model should be reevaluated and appropriately discussed.

$f(x, r)$ is continuous. We agree that Fig. 2 was confusing. Thank you for pointing this out. We remedied Fig. 2 so that it shows a continuous vector field.

Moreover, the presence of two stable equilibria in cases where “ r ” is less than zero ($r < 0$) needs further clarification or justification.

First, we added the following clarification sentence in the first paragraph of section 3.1:

“In other words, $dx/dt = f(x, r)$ with $r < 0$ has a stable equilibrium $x^* = -\sqrt{-r}$ and an unstable equilibrium $x^* = \sqrt{-r}$, which collide at $x^* = 0$ when $r = 0$.”.

Second, we added the following text five lines below in the same paragraph:

“This assumption in combination with the continuity assumption for $f(x, r)$ also implies that the stable equilibrium apart from $x^* = -\sqrt{-r}$ persists for some $r < 0$ although its position changes from $x = c$ in general. Therefore, there are two stable equilibria at least in some range of $x < 0$ near $x = 0$, as shown in Fig. 2.”.

Third, we moved up the sentence introducing Fig. 2 within the same paragraph to better assist the added sentence.

4. (minor) The authors claim that it is necessary to sample the system with at least two values of the bifurcation parameter. What are the real-world implications of this requirement and its impact on the results? How are these two values typically chosen, and is there a preference for selecting values that are distant from each other or proximate to the bifurcation point?

(i) To answer the reviewer’s first question “What are the real-world implications of this requirement and its impact on the results?”, we carried out a sensitivity analysis in which we varied the two bifurcation parameter values in the entire range of the bifurcation parameter used in our analysis. Both for the 2-node and 3-node networks, the results are very robust (in the sense that the node set S maximizing d stays the same even if we change the two values of the bifurcation parameter, r) with one exception. Even in the single exceptional case, the results are still reasonably robust against some variation in the r values. We described these new results and analysis in the new section S2 and mentioned them in the main text.

For larger networks, we set the two values of the bifurcation parameter (i.e., u or D), denoted by r with a slight abuse of notation for the sake of the discussion here, to $r^{(1)} \approx 0.1(r_{\max} - r_{\min}) + r_{\min}$ and $r^{(2)} \approx 0.9(r_{\max} - r_{\min}) + r_{\min}$, where r_{\max} and r_{\min} are the largest and smallest values, respectively, of the bifurcation parameter before the tipping point, and the approximation (i.e., \approx) only originates from the rounding. In other words, $r^{(1)}$ and $r^{(2)}$ are 10 and 90 percentiles. We did so because, if one r value is far from the tipping point and the other r value is close to the tipping point, we expect that the difference between μ_1 and μ_2 , which is the numerator of d (see Eq. (15)), is larger than the case in which the two r values are closer. Then, the difference in the d value for different choices of S may be larger, potentially making it easier to find the maximizer of d .

To test the robustness of our results with respect to the choice of $r^{(1)}$ and $r^{(2)}$, we carried out a robustness test for all the networks and all the dynamics and the two bifurcation parameters (i.e., u and D). Specifically, we set $r^{(1)}$ and $r^{(2)}$ to 10, 30, 50, 70, or 90 percentiles of the range of the bifurcation parameter values used, under the constraint that $r^{(1)} < r^{(2)}$. Then, there are nine pairs of $r^{(1)}$ and $r^{(2)}$ except the original one (i.e., $r^{(1)}$ and $r^{(2)}$ are the 10 and 90 percentiles, respectively). We measured the performance of the node set S maximizing d in terms of p_1 and p_2 . We found that the results are sufficiently robust in that p_1 and p_2 values stayed sufficiently below 1 when they did so for the original r pairs (i.e., $r^{(1)}$ and $r^{(2)}$ are the 10 and 90 percentiles, respectively) unless $r^{(1)}$ and $r^{(2)}$ are too close and $r^{(2)}$ is far from the tipping point (specifically, unless the 10%–30%, 30%–50%, 50%–70%, 10%–50%, or 30%–70% pair is used). We added a paragraph in the robustness subsection in the Results section (i.e., the paragraph starting with “Our choice of” in section 4.3) to discuss these results briefly and included the detailed setup and all the numerical results in a new SI section (i.e., section S2).

(ii) To answer the reviewer’s second question “How are these two values typically chosen, and is there a preference for selecting values that are distant from each other or proximate to the bifurcation point?”, we stated in the same added paragraph (i.e., the paragraph starting with “Our choice of” in section 4.3) the aforementioned intuition (see (i) just above) underlying our choice of the original $r^{(1)}$ and $r^{(2)}$ values. We also mentioned that our choice is regardless of the type of bifurcation. We added the intuitive explanation in this section rather than the section of the two-node or three-node network earlier because we consider that having all the discussion of the robustness of the results in section 4.3 is coherent and makes the flow of the text smoother.

5. (minor) A clear definition of what constitutes a “small” and “large” network should be given. Is it in terms of the computational complexity? Are both the number of nodes and the number edges important for calling a network “large”?

It is in terms of the number of nodes, not the number of edges. The distinction between small and large is whether or not the analytical computation is possible. To clarify this, we amended the text as follows:

- We changed the last sentence in the introduction section from
“analytically solvable small networks and with numerically investigated larger networks”
to
“analytically solvable networks with two or three nodes and with numerically investigated larger networks”.
- We changed the heading of section 3 from
“Coupled nonlinear dynamics on small networks”
to
“Coupled nonlinear dynamics on networks with two or three nodes”.
- We changed the title of Fig. 1 from
“Schematic of small networks.”
to
“Schematic of two networks.”.
- In the third paragraph in the discussion section, we changed
“systems of small size”
to
“systems with a small number of nodes”.
- We changed the text in the first sentence of section 4 from
“in larger networks, we carry out numerical simulations”
to
“in networks with a large number of nodes, we carry out numerical simulations”.

6. (minor) Section 4 needs to be reorganized to improve clarity. For instance, it could be divided into three subsections to present the results of each dynamical system separately.

We divided section 4 into four subsections. To accommodate this reorganization, we moved the last paragraph in section 4 in the previous version of the manuscript to the third subsection in the revised manuscript. We also changed the order of the relevant SI sections, which we did not mark in blue (addition) or red (deletion) in the change-tracked pdf because this is only the order change and in the SI.

Finally, apart from responding to the reviewer’s comments, we also implemented the following changes for better clarity.

- We corrected a typo in the fourth paragraph of Introduction (“an weighted” → “a weighted”) and in Eq. (1) (to add dt), and the volume number of a citation (Scheffer et al. Science 2012) in the reference list.
- We changed the range of the uniform density of the noise strength, σ_i , when it is assumed to be heterogeneous across the nodes, from $[0, 2\sigma]$ to $[0.1\sigma, 1.9\sigma]$, where σ is the mean value. This is because, with $[0, 2\sigma]$, we occasionally have noise with extremely small fluctuation (i.e., very small $\sigma_i (< 0.1\sigma)$), which is probably not realistic. This change did not modify the results substantially.
- The performance measures of the proposed node set, p_1 and p_2 , were only defined for the case in which the nodes transit from their lower to upper state as the bifurcation parameter changes in the previous version of the manuscript. In fact, the nodes are assumed to transit from the upper to the lower state as the bifurcation parameters of the mutualistic interaction dynamics and gene regulatory dynamics decrease, corresponding to, e.g., species loss as the environment worsens. Therefore, we supplied the definition of p_1 and p_2 for this case in the new section S5 and referred it from the main text. We also revised some text in the same paragraph in the main text to enhance clarity.
- We found minor errors in the code for calculating mutualistic interaction dynamics and also p_2 in some cases. Therefore, we fixed them. The results (i.e., figures) changed little.
- We corrected a few figure labels (changes not tracked because they are in the figures).
- We slightly simplified the convention of rounding for better clarity (lines 553–554), without influencing the results.
- We changed u_k to \bar{u}_k in the Methods section to avoid a notational conflict.

Response to Reviewer #2

This paper develops a powerful advance in the detection of early warning signals in networked systems. The authors present an optimization method to determine which nodes of a network are the best ones to monitor in order to extract reliable early warning signals based on phenomena such as critical slowing down. Of especial value is the finding that the method excels despite heterogeneity among nodes in terms of dynamical noise and stress. The reason this is so important is that many important systems are characterized by this heterogeneity among nodes—almost to the point that such systems are sometimes not even recognized as networks. Examples include differing countries or different climate subsystems systems that are strongly linked and subject to tipping point dynamics. Few existing methodologies provide guidance on how/which nodes to monitor with the generality that their analytical framework offers. The effect as measured by p_1 and p_2 seems very strong, and they assessed the method across a range of different networks.

The methodology is sound and the writing and organization are good.

We are glad to hear a very positive evaluation by the reviewer.

The only recommendation I have is that the authors should present some results that will more clearly demonstrate the value of the approach to less specialized readers. For example, they demonstrate the superiority of their approach by generating plots for p_1 and p_2 . This is a perfectly valid and correct method for assessing the approach. However, they should also have some plots that simply show variance versus time for the set S , compared to some random sample of nodes, for the BA model for example. This will better convey the usefulness of the approach to less mathematical audiences. Some of the important but confirmatory plots could be moved to the SI appendix to make way for such plots that can speak to a general readership.

Thanks for a great suggestion. We added a figure (Fig. 5 in the revised manuscript) which simply shows the variance versus u for the optimized node set and a uniformly randomly selected node set. We supplied text to explain this figure to help less specialized readers.

According to the reviewer's suggestion, we also moved Fig. 9 in the previous manuscript, which is a confirmatory plot, to the SI (Fig. S33 in the revised manuscript) and adjusted the text.

Apart from responding to the reviewer's comments, we also implemented the following changes for better clarity.

- We corrected a typo in the fourth paragraph of Introduction (“an weighted” → “a weighted”) and in Eq. (1) (to add dt), and the volume number of a citation (Scheffer et al. Science 2012) in the reference list.

- We changed the range of the uniform density of the noise strength, σ_i , when it is assumed to be heterogeneous across the nodes, from $[0, 2\sigma]$ to $[0.1\sigma, 1.9\sigma]$, where σ is the mean value. This is because, with $[0, 2\sigma]$, we occasionally have noise with extremely small fluctuation (i.e., very small σ_i ($< 0.1\sigma$)), which is probably not realistic. This change did not modify the results substantially.
- The performance measures of the proposed node set, p_1 and p_2 , were only defined for the case in which the nodes transit from their lower to upper state as the bifurcation parameter changes in the previous version of the manuscript. In fact, the nodes are assumed to transit from the upper to the lower state as the bifurcation parameters of the mutualistic interaction dynamics and gene regulatory dynamics decrease, corresponding to, e.g., species loss as the environment worsens. Therefore, we supplied the definition of p_1 and p_2 for this case in the new section S5 and referred it from the main text. We also revised some text in the same paragraph in the main text to enhance clarity.
- We found minor error in the code for calculating mutualistic interaction dynamics and also p_2 in some cases. Therefore, we fixed it. The results (i.e., figures) changed little.
- We corrected a few figure labels (changes not tracked because they are in the figures).
- We slightly simplified the convention of rounding for better clarity (lines 553–554), without influencing the results.
- We changed u_k to \bar{u}_k in the Methods section to avoid a notational conflict.

Reviewer #3

In the article “Anticipating regime shifts by mixing early warning signals from different nodes” by N Masuda et al., the authors devise an early warning signal for networks by optimizing the node set using a distance measure of distributions at two different bifurcation parameter values. They also compare their studies with other methods of optimizing node sets for calculating early warning signals of regime shifts. They also validate their approach to combinations of varied networks, stress, and dynamical noise.

I find the work exciting and intriguing to a broad audience, as they deal with developing optimized early warning signals for networks that are of broad applicability to regime shift prediction across diverse domains. The manuscript will add to the existing literature on early warning signals of networks, with a lot of room for expansion within the present framework. However, I have a few comments and suggestions that I strongly feel must be addressed before the manuscript can be accepted for publication.

We are glad to hear an overall very positive evaluation. We amended the manuscript according to the reviewer’s valuable comments as follows.

Below are my comments:

1. In Fig. 3, the sample covariance is distributed over a very small interval of the order 10^{-2} , whereas in Fig. 4(b), the sample covariance is distributed over a larger interval. While this can be attributed to significant noise in the later case, one should remember that in empirical data, where noise cannot be differentiated from data, such an inference could result in erroneous signals. This could be voided by measuring the change in covariance at lag one and using the normalized form of the same as an indicator of regime shift. Testing the change in covariance at higher lags could also improve early warning signals.

We totally agree with the reviewer that noise cannot be differentiated from data in empirical data. However, we are not doing any inference that depends on the noise amplitude information. While the sample covariances shown in Fig. 4(b) are indeed one magnitude (i.e., about 10 times) larger than those shown in Figs. 3, 4(a), and 4(c), we never compare across these figures when inferring anything, and the inference protocol does not change depending on the magnitude of the observed fluctuation or x_i or the intrinsic noise (i.e., σ_i). Always, we only measure the sample covariance matrix at two bifurcation parameter values (corresponding to two different network states in the case of empirical data; please see our response to item “3.” below for the application of our method to empirical data), calculate d , and thus determine the optimal node set S . Therefore, the possibility of erroneous inference along the line the reviewer points out is excluded. In the new second paragraph in the Discussion section (please also see our response to “3.” below), we articulated that our method does not use any information about the network structure or the dynamical equation models

such that it is readily applicable to multivariate time series empirical data. This implies that the method does not require any information about the dynamical noise (because the dynamical noise is part of the dynamics model in the present framework).

We also agree that measuring the variance or covariance at lag one may improve the performance of early warning signals. We had already discussed this in the previous version of the manuscript as follows (at the end of the third paragraph in the Discussion section in the revised manuscript):

“An alternative strategy is to use lagged autocorrelation as early warning signal because nodes with large intrinsic noise may produce small autocorrelation. Autocorrelation of multivariate OU processes is analytically tractable, whereas it is more complicated than the variance and covariance [Gardiner2009]. Analysis of autocorrelation and its average over nodes as early warning signals with the present theoretical framework warrants future work.”

Given that there is no problem of erroneous inference (please see our discussion just above), we prefer to keep the analysis of lagged variance/covariance/correlation for future work, as the aforementioned text states. Indeed, the analytical solutions for lagged correlation would be substantially complicated than the case of variance (e.g., around p.107 of Gardiner’s book in 2008 which we cited), and we strongly believe that it deserves another paper.

2. Secondly, the authors measure the distance d at two different values of the bifurcation parameter (r) for constructing the set S of optimized nodes. The bifurcation theory behind the occurrence of a tipping point makes it evident that changes occur in the state of the system in the vicinity of the tipping. In this study, I understand that the authors select two r values from the bistable region. How do the authors choose the different values of r to measure d in the instances of other bifurcation? I recommend the authors make it more clear whether they choose it by trial or there is some intuition at what values of the bifurcation they calculate the difference in distribution d . Also, I suspect the sensitivity of the d value on the choice of r values. A sensitivity analysis showing the effect of choice of r value on the construction of the set S theoretically or numerically would add to the novelty of the study.

The reviewer’s questions here are (i) how we are choosing the two r values in different bifurcation scenarios and intuitions behind our choice, and (ii) sensitivity analysis. Let us answer (ii) first and then (i).

(ii) Sensitivity analysis

We carried out a robustness analysis in which we varied the two r values in the entire range of the r values used in our analysis. Both for the 2-node and 3-node networks, the results are very robust (in the sense that the node set S maximizing d stays the same even if we change the r values) with one exception. Even in the single exceptional case, the results are still reasonably robust against some variation in the r values. We described these new results and the analysis methods in the new section S2 and mentioned them in the main text.

For larger networks, we set the two values of the bifurcation parameter (i.e., u or D), denoted by r with a slight abuse of notation to ease the explanation here, to $r^{(1)} \approx 0.1(r_{\max} - r_{\min}) + r_{\min}$ and $r^{(2)} \approx 0.9(r_{\max} - r_{\min}) + r_{\min}$, where r_{\max} and r_{\min} are the largest and smallest values, respectively, of the bifurcation parameter, and the approximation (i.e., \approx) only originates from the rounding. In other words, $r^{(1)}$ and $r^{(2)}$ are 10 and 90 percentiles. We did so because, if one r value is far from the tipping point and the other r value is close to the tipping point, we expect that the difference between μ_1 and μ_2 , which is the numerator of d (see Eq. (15)), is larger than the case in which the two r values are closer. Therefore, the difference in the d value for different choices of S may be larger, potentially making it easier to find the S maximizing d .

To test the robustness of our results with respect to the choice of $r^{(1)}$ and $r^{(2)}$, we carried out a robustness test for all the networks and all the dynamics and the two bifurcation parameters (i.e., u and D). Specifically, we set $r^{(1)}$ and $r^{(2)}$ to 10, 30, 50, 70, or 90 percentiles of the range of the bifurcation parameter values used under the constraint that $r^{(1)} < r^{(2)}$. There are nine pairs of $r^{(1)}$ and $r^{(2)}$ except the original one (i.e., $r^{(1)}$ and $r^{(2)}$ are the 10 and 90 percentiles, respectively). We measured the performance of the node set S maximizing d in terms of p_1 and p_2 . We found that the results are sufficiently robust in that p_1 and p_2 values stayed sufficiently below 1 when they did so for the original r pair unless $r^{(1)}$ and $r^{(2)}$ are too close and $r^{(2)}$ is too far from the tipping point (i.e., unless the 10%–30%, 30%–50%, 50%–70%, 10%–50%, or 30%–70% pair is used). We added a paragraph in the robustness subsection in the Results section (i.e., the paragraph starting with “Our choice of” in section 4.3) to discuss these results briefly and included the detailed setup and all the numerical results in section S2.

Please also note that our theory does not depend on the $r^{(1)}$ and $r^{(2)}$ values, and therefore we decided to run this robustness test only numerically.

(i) Intuitions behind our choice of the r values

In the same new paragraph in section 4.3, we also stated the aforementioned intuition underlying our choice of the original $r^{(1)}$ and $r^{(2)}$ values. We also mentioned that our choice is regardless of the type of bifurcation. We added the intuitive explanation in this section rather than the section of the two-node or three-node network earlier because we consider that having all the discussion of the robustness of the results in section 4.3 is coherent and makes the flow of the text smoother.

3. Additionally, it is important to note that the entire significance of early warning signals is directed to their utility in anticipating regime shifts in real data. Throwing some light on how one can extend this study to empirical data will add to the generality of the study, which is not apparent in the present form of the manuscript.

The proposed methods are directly applicable to real data because it does not require the knowledge of the network structure or the dynamical system equations. We added a paragraph in the Discussion section (i.e., the second paragraph in the Discussion section) to explain how to apply our methods to empirical data, citing example data from ecology and psychopathology.

4. The study of early warning signals here is only focused on trends obtained from data undergoing a regime shift, and their strength is measured using Kendall's tau. However, it is essential to assess the chance of false negatives (i.e., signals in data not undergoing a regime shift) as well. Clearly, previous literature has shown evidence of a rise in variance and artificial autocorrelation in data not undergoing regime shift. Therefore, it is necessary to perform an analysis of the early warning signals in data not undergoing regime shifts and investigate the strength of signals in the same. This is absent in the present study.

Here are some important references that the authors might find relevant.

Boettiger C, Hastings A. Early warning signals and the prosecutor's fallacy. *Proceedings of the Royal Society B: Biological Sciences*. 2012 Dec 7;27 (1748):4734-9.

Boettiger C, Ross N, Hastings A. Early warning signals: the charted and uncharted territories. *Theoretical ecology*. 2013 Aug;6:255-64.

Thanks for constructive comments and giving us important references. In the Venn diagram in Fig. 1 of Boettiger et al. *Theor. Ecol.* (2013), the false positive cases that the reviewer mentioned correspond to regions III and IV. A major scenario of this case is transcritical bifurcations, shown in region III. Under transcritical (or Hopf) bifurcations, a (large) regime shift is not observed while the critical slowing down occurs, so that traditional early warning signals would increase (which is a false positive). In fact, we did these cases already in the previous version of the manuscript; the gene regulatory model and the SIS model, which are two of the four dynamics models used in our study, show transcritical bifurcations (at least in the case of a single node and well-mixed populations). In the previous version of the manuscript, we showed that a rise in our early warning signal, i.e., \hat{V}_S , is observed through the transcritical bifurcations in these models, quantified as large Kendall's τ values (shown in old Fig.9, which is new Fig.S33; also see new Fig.S34–S44). Therefore, our methods do yield false positives, and this is consistent with previous studies (e.g., Kefi et al., *Oikos* (2012), which is cited as a reference for region III in Fig.1 of Boettiger et al. (2013)). What was missing in the previous version of the manuscript was the discussion of this, including that the gene regulatory model and SIS models show transcritical bifurcations. Therefore, we added a paragraph in the new robustness subsection in the Results section to discuss false positives (the second last paragraph in section 4.3).

Because the reviewer mentioned the importance of false negatives, we also carried out additional numerical experiments to assess what occurs in the case of false negatives. This corresponds to regions II and V in Fig. 1 of Boettiger et al. (2013) because, in these regions, a large regime shift occurs without accompanying critical transitions. Among several scenarios in which this occurs, we focused on two main scenarios (Boettiger et al. (2013); Scheffer et al., *Nature* (2009); Scheffer et al., *Science* (2012)), i.e., regime shifts driven by dynamical noise and those driven by large impulse input. Importantly, neither of these two scenarios accompany changes in the system parameter values. We showed

that a rise in our early warning signals, \hat{V}_S , was not observed for any node set S in these two scenarios, except very near the regime shift. We discussed these results in a new paragraph added to section 4.3 (i.e., the paragraph starting with “Not all the regime shifts”) and showed the details in the new section S7.

We also cited these two important references by Boettiger and colleagues in the introduction and results section. We also cited Hastings & Wysham, *Ecology Letters*, 13, 464–472 (2010) in our discussion of false negatives in section 4.3 (i.e., the paragraph starting with “Not all the regime shifts”). We also changed “near a tipping point” to “near such a tipping point” in the first paragraph of the Introduction section to indicate that not all tipping points accompany critical slowing down.

5. In Fig. 5, the authors show that increasing the size of N improves the early warning. However, in another instance, the authors mention, “The node set maximizing d tends to work better for smaller node sets (i.e., smaller n).” This may be made clear.

The latter sentence was confusing and inaccurate. We rewrote this sentence as follows:

“The node set maximizing d tends to work better relative to uniformly random node sets with the same number of nodes, n , when n is smaller.”

For the same reason, we replaced

“The node set maximizing d tends to work better (i.e., smaller p_1 and p_2 values) when ...”

in the same paragraph by

“The node set maximizing d tends to work better relative to uniformly randomly picked node sets, yielding smaller p_1 and p_2 values, when ...”.

Similarly, we replaced

“The performance degrades as n increases.”

in the last paragraph of section 4.2 in the revised manuscript by

“The performance of the optimized node set relative to that of uniformly randomly selected node sets with the same number of nodes, n , degrades as n increases.”.

6. I find that a limitation of the algorithm is that there are no stopping criteria in constructing S . I think the authors may find it interesting to derive stopping criteria in the construction of the set S as increasing the size of N improves the signal, and this can lead one to continue calculating until one reaches the size of the network. This may be computationally very expensive and undermines the utility of the same when working with higher dimensional real networks.

We devised a stopping criterion and carried out a numerical demonstration. We briefly mentioned the method and results, including the motivation, in the main text (lines 272–273 and 455–457) and provided the details, including the numerical results, in the new section S4. We also provided code to maximize d with the proposed stopping criterion (please see section S4).

Finally, apart from responding to the reviewer’s comments, we also implemented the following changes for better clarity.

- We corrected a typo in the fourth paragraph of Introduction (“an weighted” → “a weighted”) and in Eq. (1) (to add dt), and the volume number of a citation (Scheffer et al. Science 2012) in the reference list.
- We changed the range of the uniform density of the noise strength, σ_i , when it is assumed to be heterogeneous across the nodes, from $[0, 2\sigma]$ to $[0.1\sigma, 1.9\sigma]$, where σ is the mean value. This is because, with $[0, 2\sigma]$, we occasionally have noise with extremely small fluctuation (i.e., very small $\sigma_i (< 0.1\sigma)$), which is probably not realistic. This change did not modify the results substantially.
- The performance measures of the proposed node set, p_1 and p_2 , were only defined for the case in which the nodes transit from their lower to upper state as the bifurcation parameter changes in the previous version of the manuscript. In fact, the nodes are assumed to transit from the upper to the lower state as the bifurcation parameters of the mutualistic interaction dynamics and gene regulatory dynamics decrease, corresponding to, e.g., species loss as the environment worsens. Therefore, we supplied the definition of p_1 and p_2 for this case in the new section S5 and referred it from the main text. We also revised some text in the same paragraph in the main text to enhance clarity.
- We found minor error in the code for calculating mutualistic interaction dynamics and also p_2 in some cases. Therefore, we fixed it. The results (i.e., figures) changed little.
- We corrected a few figure labels (changes not tracked because they are in the figures).
- We slightly simplified the convention of rounding for better clarity (lines 553–554), without influencing the results.
- We changed u_k to \bar{u}_k in the Methods section to avoid a notational conflict.

REVIEWERS' COMMENTS

Reviewer #1 (Remarks to the Author):

The authors did an excellent job to address my comments as well as those of the other two referees, resulting in a significantly improved manuscript. I recommend it for Nature Communications.

Reviewer #2 (Remarks to the Author):

The authors have responded to my comments in a satisfactory way.

Reviewer #3 (Remarks to the Author):

The authors have addressed all the issues I raised, and I think the paper can now be accepted for publication in Nature Communications.